# Stable isotope profiles of soil organic carbon in forested and grassland landscapes in the Lake Alaotra basin (Madagascar): insights in past vegetation changes.

Vao Fenotiana Razanamahandry[1], Marjolein Dewaele[1], Gerard Govers[1], Liesa Brosens[1,2], Benjamin Campforts[3], Liesbet Jacobs[1,4], Tantely Razafimbelo[5], Tovonarivo Rafolisy[5] and Steven Bouillon[1].

[1]Departement of Earth and Environmental Sciences, KU Leuven, Leuven, Belgium
[2]Research Foundation Flanders (FWO), Egmontstraat 5, 1000 Brussels, Belgium
[3]Institute for Arctic and Alpine Research, University of Colorado at Boulder, Boulder, CO, USA
[4]Institute for Biodiversity and Ecosystem Dynamics, University of Amsterdam, Amsterdam, Netherlands
[5]Laboratoire des Radio-Isotopes, Université d'Antananarivo, Antananarivo, Madagascar

*Correspondence to*: Vao Fenotiana Razanamahandry (vaofenotiana.razanamahandry@kuleuven.be)

**Abstract.** The extent to which the central highlands of Madagascar were once covered by forests is still a matter of debate: while reconstructing past environments is inherently difficult, the debate is further hampered by the fact that the evidence documenting land cover changes and their effects on carbon and sediment dynamics in Madagascar has hitherto mainly been derived from lake coring studies. Such studies provide an integrated view over relatively large areas but do not provide information on how land use change affects hillslopes in terms of carbon and sediment dynamics. Such information would not only be complementary to lake inventories but may also help to correctly interpret lake sediment data. Carbon stable isotope ratios ($\delta^{13}$C) are particularly useful tracers to study the past dynamics of soil carbon over timespans ranging from years to millennia and thus to understand the consequences of land-use change over such timespans. We analyzed soil profiles down to a depth of 2 m from pristine forests and grasslands in the Lake Alaotra region in central Madagascar. Along grassland hillslopes, soil organic carbon (SOC) content was low, from 0.4 to 1.7 % in the top layer, and decreased rapidly to ca. 0.2 % below 100 cm depth. The current vegetation predominantly consists of C4 grasses ($\delta^{13}$C ~-13 ‰), yet topsoil $\delta^{13}$C-OC range between -23.0 and -15.8 ‰, and most profiles show a decrease in $\delta^{13}$C-OC with depth. This contrasts with our observations in the C3-dominated forest profiles, which show a typical profile whereby $\delta^{13}$C values increase slightly with depth. Moreover, the SOC stock of grasslands was ~55.6 % lower than along the forested hillslopes for the upper 0–30 cm layer. $\delta^{13}$C values in grassland and forest profiles converge to similar values (within $2.0 \pm 1.8$ ‰) at depths below ~80 cm, suggesting that the grasslands in the Lake Alaotra region have indeed developed on soils formerly covered by a tree vegetation dominated by C3 plants. We also observed that the percent of modern carbon (pMC) of the bulk OC in the top, middle and lower middle positions of grasslands was less than 85 % near the surface. This could reflect a combination of (i) the long residence time of forest OC in the soil, (ii) the slow replacement rate of grassland-derived OC, (iii) and the substantial erosion of the top positions towards the valley position of grasslands. At the valley positions under grassland, the upper 80 cm contains higher amounts of recent

grass-derived OC in comparison to the hillslope positions. This is likely to be related to the higher productivity of the grassland valleys (due to higher moisture and nutrient availability), and deposition of OC that was eroded further upslope may also have contributed. The method we applied, which is based on the large difference in $\delta^{13}C$ values between the two major photosynthetic pathways (C3 and C4) in (sub)tropical terrestrial environments, provides a relatively straightforward approach to quantitatively determine changing vegetation cover, and we advocate for its broader application across Madagascar to better understand the island's vegetation history.

Keywords: carbon isotope, soil organic carbon, carbon stock, vegetation change.

## 1 Introduction

Madagascar is an island characterized by highly distinct landscapes. The eastern part of the island is humid and covered by tropical rainforest while the southern and western regions are drier and dominated by (wooded) grasslands including shrubby bushlands and arborescent species of *Euphorbia*. Madagascar is often considered to experience very high erosion rates (Szabó et al., 2015). The most spectacular erosion-forms on the island are large inverse-teardrops shaped gullies or "lavaka" meaning "hole" in Malagasy language. Lavaka are omnipresent in the central highlands of Madagascar (Brosens et al., 2022; Cox et al., 2010; Szabó et al., 2015; Voarintsoa et al., 2012; Wells and Andriamihaja, 1993) and frequently initiate on convex slopes covered by grassland vegetation and patches of shrubs (Brosens et al., 2022; Wells and Andriamihaja, 1993). Active lavaka are absent under forest vegetation (Aubréville and Bossanyi, 2015).

Cox et al. (2009) concluded from [10]Be concentrations that ~80 % of the sediment in the Malagasy highland rivers originated from lavaka and concluded that these were already widely present when humans arrived. However, a recent study by Brosens et al. (2022) shows that, although lavaka probably have indeed been present for millennia, their number strongly increased over the last 4 centuries due to a strong increase in environmental pressure exerted by an ever growing rural population. Although it is undisputed that environmental pressure has indeed increased since the arrival of humans, the question remains to what extent human activities have changed the land cover of the Malagasy highlands. The isolated residual forest patches that are present within grassland regions are believed to provide evidence that the Malagasy highlands were nearly completely forested prior to the arrival of humans (Chevalier, 1922). Baron (1889) already believed that the whole eastern part of the island was densely forested, but did not present solid evidence for this claim. Other early researchers, however, questioned whether the forest covered the whole island in the past (e.g. Battistini and Verin, 1972). They suggested that the drier parts of the island were likely covered by an alternation of grasslands and sparse forests mainly composed of tall shrubs and xerophyllous vegetation, while the more humid parts were generally covered by a dense forest. They assumed that the latter was dramatically reduced by human deforestation, resulting into a transition to grasslands and leading to more intense erosion (Chevalier, 1922). This view has been widely adopted by major international development institutions that strongly stimulated afforestation in grassland regions in Madagascar. However, Bond et al. (2008) claimed that the Malagasy grasslands have an

ancient origin that was part of the global expansion of C4 grass biomes at the end of the last glacial period, and are thus not anthropogenically driven. They conclude this based on (i) evidence that the extinct Malagasy fauna consumed C4 or CAM

(Crassulacean Acid Metabolism) plants and (ii) the diversity of C4 grass lineages in Madagascar relative to the mainland African continent. Recent work relying on floristic inventories revealed that 40 % of Madagascar's grass species are endemic, supporting the argument for an ancient origin of the open grassland vegetation on the island (Vorontsova et al., 2016). Furthermore, different centers of grass endemism were identified, suggesting that grasslands were scattered across the island before human arrival. Evidence from several sites, using dated fossil pollen, charcoal and diatoms further confirm this, showing

that graminoid vegetation was present in the Malagasy highlands before human settlement (Burney, 2004; Gasse and Van Campo, 2001; Matsumoto and Burney, 1994).

These two views are not necessarily contradictory. While it is clear that grasslands are indeed endemic to Madagascar, there is also strong evidence that the fraction of the land covered by grasses has increased over the last 1500–1000 years (Virah-Sawmy et al., 2016). Pollen profiles suggest that this "opening up" of the landscape may have occurred relatively rapidly, over

a time scale of centuries. It remains very difficult to quantify the extent of this vegetation change: while pollen diagrams from lake deposits do provide an integrated assessment, they do not provide information on where in the landscape changes may have occurred nor do they allow us to quantify the degree of change (Straka, 1996; Virah-Sawmy et al., 2016). Furthermore, lake sediment studies do not allow us to assess how vegetation changes affect carbon and sediment dynamics on the hillslopes, making it impossible to derive solid conclusions on how carbon inventories and fluxes of carbon and sediment may have

changed due to vegetation changes.

Land-use cover changes such as deforestation represent an important anthropogenic contribution to the emission of greenhouse gases in tropical regions (Harris et al., 2012), and the concurrent increase in erosion rates (Restrepo et al., 2015) can significantly impact soil organic carbon (SOC) stocks (Batlle-Bayer et al., 2010; Rabetokotany-Rarivoson et al., 2015; Yang et al., 2018). On the one hand, soil erosion induces SOC losses due to structural aggregate degradation and due to reduced

productivity resulting from a decrease in soil nutrients in the eroding area (Jacinthe et al., 2002; Lal, 2004). On the other hand, soil erosion can lead to a SOC sink in the depositional area due to transfer and burial of high SOC from the eroding area (Van Oost et al., 2007). In (sub)tropical environments, stable isotopes offer one of the possible approaches to quantify land use and vegetation changes at the landscape scale (Boutton et al., 1998). Forest vegetation (following the C3 photosynthetic pathway) and grasses (following the C4 pathway for the majority of tropical and subtropical species) show a different degree of isotope

fractionation (Peterson and Fry, 1987), and therefore, stable carbon isotope ratios ($\delta^{13}C$) of SOC have been an important technique to understand vegetation shifts ( e.g., Desjardins et al., 2013, 2020).

The main aim of this study is therefore to provide insight into the vegetation history and the impact of vegetation changes on carbon and sediment dynamics at the hillslope scale of the grasslands in the Lake Alaotra region located in the central highlands of Madagascar. As it is the most important rice-producing region in Madagascar, population density is high. Irrigated rice is

located mainly in the lake's plains while the surrounding hills and mountains are used for extensive cattle herding and, to a much smaller extent, for arable crops (Penot et al., 2018). While much of the Lake Alaotra catchment consists of grassland-

covered hills with a high density of lavaka (Cox et al., 2010; Voarintsoa et al., 2012), pristine forests are located just east of the lake. These forests form the westernmost part of a larger rainforest-covered region extending from north to south along the east coast of the island. We sampled soil profiles along hillslope transects at both forested and grassland sites whereby organic carbon (OC) stocks were quantified, and $\delta^{13}C$ values and $^{14}C$ activity of soil OC were measured to evaluate differences in OC stocks and OC isotopic signatures between grassland and forest land. The stable carbon isotope ratio ($\delta^{13}C$) is an interesting tracer to understand vegetation shifts in a landscape over time because forest plants (following the C3 photosynthetic pathway) and grasses (following the C4 pathway for the majority of tropical and subtropical species) show a different degree of isotope fractionation (Cerling and Harris, 1999). This results in C3 plants having more negative $\delta^{13}C$ values with an average of approximately -28 ‰, while C4 plants have less negative $\delta^{13}C$ values around -13 ‰ (Peterson and Fry, 1987). $^{14}C$ activity measurements of the bulk SOC, on the other hand, allow us to assess the (relative) age of SOC. We use this information to assess whether the current grassland sites were formerly forested and how vegetation cover changes may have affected carbon and sediment dynamics on the hillslopes.

## 2 Materials and methods

### 2.1 Study area: Lake Alaotra region

Lake Alaotra is the largest lake of Madagascar, located in the central eastern part (17–18° S and 48–49° E) at about 775 m above sea level (a.s.l) (Mietton et al., 2018) (Figure 1a). The surrounding hills rise up to 900–1300 m a.s.l. (Bakoariniaina et al., 2006). Precambrian crystalline rocks, predominantly magmatic rocks of the graphitic system of Madagascar, underlie the area. Lake Alaotra is located in a graben bounded by faults on its eastern and western sides that probably originated in the Tertiary in response to Neogene extensional tectonic events (Mietton et al., 2018). The rainfall (mean annual precipitation between 900 and 1250 mm) is characterized by a distinct wet and dry season with the occurrence of tropical cyclones. The mean annual temperature is 20.6 °C, ranging from 11°C in July to 28 °C in January (Ferry, 2009).

Nowadays, most of the hills surrounding Lake Alaotra are covered by C4 grasses. These areas are being used for cattle herding and are characterized by a high density of lavaka (Voarintsoa et al., 2012). Recent research has shown that lavaka in the Lake Alaotra region (Supplementary figure S1) are on average ca. 400 years old. Lavaka became far more numerous since ca. 1000 years ago and lavaka formation rates have increased dramatically over the last 200 years. This timing and the rapid increase in lavaka erosion rates has been confirmed by floodplain sedimentation data in the same area (Brosens et al., 2022). Brosens et al. (2022) link this increase in lavaka erosion to increased environmental pressure due to growing human populations and intensified grazing based on scenario modelling and on the absence of significant climatic variations in this period.

The soils in our study area are classified as ferralsols according to the World Reference Base for Soil Resources (2006). They are underlain by a deeply weathered regolith that is in most locations several tens of meters thick. At many locations, a hard lateritic horizon has developed in the upper part of the regolith, covering it like a "shield" (Voarintsoa et al., 2012; Wells and Andriamihaja, 1993). The lateritic soil horizon is usually between 0.5 to 2 m thick (Voarintsoa et al., 2012), and is considered

to be relatively impermeable, thus favouring surface runoff, especially if there is no or little vegetation and if no cracks are
130 present (Wells and Andriamihaja, 1993). In addition, the basement rocks on the site are metamorphic and igneous (Du Puy and Moat, 1996). Therefore, geogenic OC is not considered to be substantial, in contrast to subsoils developed from sedimentary rocks where this might be more important (Graz et al., 2010).

While most of the Lake Alaotra region is covered by C4 grasses, a small part of the eastern side of the catchment is covered by forest which is part of the Zahamena National Park. Zahamena is the third strict Nature Reserve of Madagascar that was
135 established in 1927 and since then no human occupancy has been allowed within its boundaries except for scientific research (Andriamampianina, 1984). This National Park contains important remnants of the evergreen rainforest of Madagascar at low (0–800 m) and medium altitudes (800–1800 m) (Styger et al., 2007) and extends over 3710 km$^2$ (Andriamananjara et al., 2016) (Figure available online, Supplementary Material S2). The rainforest of the Zahamena National Park is one of the most diverse in Madagascar with many species of flora with a high degree of endemism (Raboanarielina, 2012). This pristine forest zone is
140 part of the Eastern Forest corridor of the East of Madagascar (Figure 1), where Hackel et al. (2018) have inventoried endemic C3 grass species in Madagascar. These endemic C3 grasses belong to the group "forest shade clade" (Paniceae: Boivinellinea) and their diversification since the Miocene is reported to be favoured by the expansion of the Sambirano rainforest (in the North of Madagascar) (Hackel et al., 2018; Yoder and Nowak, 2006).

## 2.2 Sampling transects

The hillslopes surrounding Lake Alaotra are nearly all convex-sloping hills (Supplementary Material S3), a characteristic that is commonly observed throughout the central highlands of Madagascar (Wells and Andriamihaja, 1993). These convex hillslopes typically have a rounded flat top and steepen downslope with the highest slope gradients near the bottom of the hillslope (Figure 1b and 1c; Figures available online: Supplementary Material S3 and S4).

Vegetation varies with topography: grass vegetative cover is typically highest in the valleys and lowest at the hillslope
convexities (Figure available online: Supplementary Material S2 and S4).

Soil samples were collected in the lateritic horizon, along four convex hillslope transects, two of which are located in the grassland habitat, while two hillslopes under pristine forest were sampled just east of the basin boundary in the Zahamena Forest Corridor (Figure 1). The two chosen grassland hillslopes are denoted as GLP (Grassland hillslope where Lavaka is Present) and GLA (Grassland hillslope where Lavaka is Absent). For GLP, the chosen transect was situated outside the actual
lavaka. By choosing a slope with and without lavaka, we wanted to investigate whether soils on slopes that have lavaka development may differ from slopes that do not have them. The two pristine forest transects are denoted as F1 and F2. The slope gradients (derived from the 12 m resolution TanDEM-X DEM) of the four transects were comparable, with the maximum slope gradient of 25° and 30° for the forest transect and 25° and 29° under grassland (see Supplementary Figure S3).

Along each of the hillslope transects, six soil profiles were sampled down to 2 meters deep. We named the sampling positions
top (T), upper-middle (UM), middle (M), lower-middle (LM), bottom (B) and valley (V), respectively (Figure 1 and Figure available online: Supplementary Material S3). Soil samples were collected using a motorized auger with 10 cm diameter for

the upper 0–100 cm and 7.5 cm for the soil layer between 100 and 200 cm. The sample resolution was 5 cm for the first meter and 10 cm for the second meter. Compaction of the soil during sampling was negligible given the wide auger diameters. Soil samples were taken using Kopecky rings (diameter = 4 cm and height = 4 cm) in the open side of the auger. In total, 720 soil samples were collected, 180 per transect and 30 per sampling location. We additionally collected leaves of the most abundant vegetation types both at the grassland and forest sampling sites.

### 2.3 Laboratory and data analyses

#### 2.3.1 OC content, $\delta^{13}C$ and $^{14}C$ measurements

Soil and vegetation samples were first air-dried and later dried in an oven at 50 °C for several days. Soil samples were homogenized with a mortar and pestle. Liquid nitrogen was used while crushing the vegetation samples to facilitate their homogenization and vegetation subsamples were placed in tin cups (8 x 5 mm). For each soil sample, a sub-sample was weighed into 8x5 mm Ag cups and acidified by adding a 40 µL at HCl (10 %) solution to eliminate inorganic C. The methodology and efficacy of this acidification method are described in detail elsewhere (Kennedy et al., 2005; Komada et al., 2008).

Both soil and vegetation sub-samples were analysed using an elemental analyser-isotope ratio mass spectrometer (Thermo Flash HT/EA or CE 1110 coupled to a Delta V Advantage) to determine the OC content and $\delta^{13}C$ of soil OC and vegetation samples. Data calibrations were performed using multiple analyses of different standards within each run: caffeine (IAEA-600) and two in-house reference materials (leucine and tuna muscle tissue) which were calibrated versus certified standards. Reproducibility of $\delta^{13}C$ measurements was better than ± 0.2 ‰. Carbon isotope ratios are expressed in the δ-notation:

$$\delta^{13}C = ([R_{sample} - R_{std}]/R_{std}) \times 10^3 , \tag{1}$$

where $R_{sample}$ is the $^{13}C/^{12}C$ ratio of the sample and $R_{std}$ is the $^{13}C/^{12}C$ ratio of the international standard V-PDB (Vienna Pee Dee Belemnite) (Hoefs, 2009). In principle, the contribution of C4-vegetation derived carbon to the OC pool at grassland sites can be estimated by applying a simple mixing model:

$$\%C4_{gs,i,j} = [(\delta^{13}C_{s,i,j} - \delta^{13}C_f)/ (\delta^{13}C_g - \delta^{13}C_f)] \times 100 , \tag{2}$$

where $\%C4_{gs,i,j}$ is the contribution of the C4 vegetation in the SOC expressed in % in grassland profile $i$ at depth $j$; $\delta^{13}C_{s,i,j}$ is the isotope ratio of the soil sample in grassland profile $i$ at depth $j$, $\delta^{13}C_g$ is the isotope ratio of grassland vegetation and $\delta^{13}C_f$ is the isotope ratio of forest vegetation.

However, this mixing model implicitly assumes that the $\delta^{13}C$ ratio in a forest soil equals the $\delta^{13}C$ ratio in C3 vegetation, which is not the case: $\delta^{13}C$ values in soils under C3 vegetation tend to systematically deviate from those observed in near-ground vegetation and these deviations vary with depth. Such variations can be accounted for by using a modified form of Eq. (2) whereby the $\delta^{13}C$ signal measured under forest at the same depth is used as the reference rather than the $\delta^{13}C$ of the C3 vegetation. Equation (2) then becomes:

$$\%C4_{gs,i,j} = [(\delta^{13}C_{s,i,j} - \delta^{13}C_{fs,i,j})/ (\delta^{13}C_g - \delta^{13}C_{fs,i,j})] \times 100 , \tag{3}$$

where $\delta^{13}C_{fs,i,j}$ is the average $\delta^{13}C$ value of a soil sample taken under the C3 forest vegetation at the same depth.

For soils that have permanently been under C4 vegetation, evidence suggests that the difference between C4 values in soils and corresponding vegetation is relatively limited (Conrad et al., 2017; Desjardins et al., 2020) and that the variations of $\delta^{13}C$ with depth are limited under C4 vegetation.

Here we used values ranging from -15 to -13 ‰ as the reference values for soils with C4 vegetation. These values are similar to the values observed by Conrad et al. (2017) in soils under permanent C4 vegetation in a research station in Gayndah, in southern Queensland, Australia. We then obtain:

$$\%C4_{gs,i,j} = [(\delta^{13}C_{s,i,j} - \delta^{13}C_{fs,i,j})/ (\delta^{13}C_{g,ref} - \delta^{13}C_{fs,i,j})] \times 100 , \tag{4}$$

where $\delta^{13}C_{g,ref}$ is the reference $\delta^{13}C$, i.e. the $\delta^{13}C$ we expect for a soil profile that has permanently been under C4 vegetation (from -15 to -13 ‰). Finally, the contribution of C3 vegetation in the SOC in grassland profile $i$ at depth $j$ (expressed in %) will be:

$$\%C3_{gs,i,j} = 100 - \%C4_{gs,i,j} . \tag{5}$$

Radiocarbon analyses were performed on bulk soil organic matter of the surface soil samples (5–10 cm) for each sampling position of F2 and GLP. Additional $^{14}C$ measurements were made at 20–25 cm, 50–55 cm and 100–110 cm depth for GLP-T, GLP-V, F2-T and F2-V (see details on Supplementary Material). These measurements were performed on bulk soil samples which were acidified to remove carbonates. Sample pre-treatment and $^{14}C$ activity measurements were carried out at the Radiocarbon Laboratory of the Royal Institute of Cultural Heritage (Brussels, Belgium), using a MICADAS accelerator mass spectrometer.

In an open system such as a soil profile, there is a continuous exchange of C with the surrounding environment, and $^{14}C$ measurements of bulk soil organic C should thus be interpreted with caution since it is influenced by the inputs and the outputs of C within the soil (Trumbore, 2009; Wang et al., 1996). Thus, while $\Delta^{14}C$ data on bulk SOC do not necessarily correspond to absolute dates of the formation of the C within a given depth layer and thus do not suffice to precisely date the timing of a vegetation change, they at least provide a relative framework and shed light on how fast the SOC pool may respond in a changing environment (e.g. Desjardins et al., 2020; Krull et al., 2005; Trumbore, 2009). We used our actual $^{14}C$ measurements of bulk SOC to calculate the percent Modern Carbon (pMC), as recommended for an open system such as soil (Reimer et al., 2004; Schuur et al., 2016). The pMC reflects the incorporation of C fixed from the atmosphere since atomic weapon testing in the early 1950s, which nearly doubled the activity of $^{14}C$ in the atmosphere (Reimer et al., 2004). A pMC greater than 100 % indicates that the majority of SOC bulk was fixed post-1950s (Krull et al., 2005; Trumbore, 2009). We also provide data expressed in radiocarbon ages (years BP, see supplementary Material). For sample where pMC >100 %, no $^{14}C$ age is provided since these indicate the presence of bomb-derived carbon (post-1950's) (Krull et al., 2005).

### 2.3.2 Dry bulk density and estimation of carbon stock

The dry bulk density ($\rho_d$) was calculated by using the dry weight of the soil samples, and the known volume of the Kopecky rings. Absolute cumulative SOC stocks were calculated by integrating the carbon amount of each depth interval, using the dry bulk density and the organic carbon concentration:

$$q(i) = \%OC(i) \times D(i) \times \rho_d(i), \tag{6}$$

$$Q(z) = \sum q(i), \tag{7}$$

where $q(i)$ is the carbon stock in soil depth interval $i$ (units converted into Mg C ha$^{-1}$), $\%OC(i)$: organic carbon content in soil depth interval $i$, $D(i)$: the thickness of horizon $i$ (cm), $\rho_d(i)$: dry bulk density of the soil depth interval $i$ (g cm$^{-3}$). The estimated absolute value of the cumulative SOC stock ($Q(z)$) is the amount of SOC stored between the soil surface and soil depth $z$ (Cerri et al., 2007).

    The relative value of the cumulative SOC stocks at depth $z$ ($R(z)$, %) was calculated as the ratio of the absolute value of the cumulative SOC stock at this depth ($Q(z)$) to the absolute value of the SOC stock for the whole soil profile (i.e. between 0 and

2 m depth, $Q$):

$$R(z) = Q(z) * 100/Q. \tag{8}$$

### 2.3.3 Soil texture analyses

    To analyse the soil texture, we pre-treated dry soil subsamples (15 to 20 g) with HCl (5 %) to remove carbonates. Subsequently,

organic matter was removed with a $H_2O_2$ (35 %) solution at 40 °C, and each soil subsample was washed with demineralized water. A peptising solution of 50 mL sodium oxalate and sodium carbonate was added to the pre-treated soil samples. Afterwards, water was added to 150 mL, the mixture was boiled for 10 minutes, and allowed to cool to room temperature. Sand (0.63–2 mm) was separated by wet sieving, and silt (2–63 µm) and clay (< 2 µm) fractions were determined by sedimentation and decantation. Material retained and collected from sieves and decantation were oven-dried and weighed.

### 2.4 Statistical analyses

    Overall comparisons between two groups were done using a student t-test if the data were normally distributed and passed an equal variance test. For non-normally distributed data, non-parametric tests (Mann-Whitney U-test) were used. Differences were considered statistically significant at $P < 0.05$. To compare more than 2 groups, standard one-way analysis of variance was performed, with parametric pairwise test (Tukey's honestly significant difference test) comparisons for normally

distributed data. Non-parametric tests using one-way analysis of variance on ranks (Kruskal-Wallis), followed by non-parametric pairwise comparison (Dunn's test) were used for non-normally distributed data. The relationship between $\delta^{13}$C and pMC of SOC was tested using ordinary Pearson correlation and least squares linear regressions. Linear regressions were used despite not having an explicit independent variable, as we sought only the presence or absence of a relationship between pMC and $\delta^{13}$C of SOC.

## 3 Results

### 3.1 Soil texture for grassland and forest soils

For the transects under forest, the proportion of clays ranged between 46.3 and 50.2 %, silt ranged between 8.5 and 15.2 %, and the sand fraction constituted between 36.1 and 43.6 % of the soil mass. For the transects under grassland, clays were the dominant fraction (39.5–52.4 %), followed by sand (28.2–46.7 %) and silt (3.0–26.9 %). There were no significant differences in texture of soils under grassland and forest (p-value = 0.7 (sand), p-value = 0.7 (silt) and p-value = 0.2 (clay)). Soil texture under both grassland and forest was thus classified as sandy clays to clays (Supplementary figure S5).

### 3.2 Organic carbon content and stable carbon isotope of forest soil and vegetation

The OC content of the forest profiles ranged between 1.5 and 4.6 % for F1 and between 2.9 and 5.6 % for F2 for the topsoil samples (0–5 cm, Figure 2a and 2b). Overall, the %OC trends of the profiles were similar for the different sampling locations, where the %OC was highest in the topsoil, decreasing exponentially with depth over the first ~60 cm. At 190 cm depth, %OC decreased to 0.1–0.3 % for both profiles. For forest transect F1, %OC content in the upper ca. 60 cm was higher at the UM and LM positions compared to the other hillslope positions. These differences were particularly apparent in the upper ca.~20 cm of the profile. This difference between the hillslope positions was less marked in F2. However, the %OC was highest for the F2-B, F2-UM and F2-LM when compared to the F2-V, F2-T and F2-M.

At the surface, the $\delta^{13}C$ values only showed minor variations, ranging between -27.1 and -25.5 ‰ (0–5 cm, Figure 3a and 3b). $\delta^{13}C$ values increased with depth and reached a value of -24.1 ± 0.6 ‰ at a depth of 60 cm. Below this depth, $\delta^{13}C$ values no longer showed a systematic variation with depth but varied within a narrow range. No distinct trends were observed for the different sampling positions, where only the $\delta^{13}C$ of F1-T were slightly higher throughout depth when compared to the other sampling positions. $\delta^{13}C$ values of profiles under forest (F1 and F2) did not differ significantly in the upper 50 cm depth of the profiles (Supplementary Table 1). The vegetation samples from the forest showed $\delta^{13}C$ values that are clearly lower than those observed in the forest soils, with values between -35.8 and -30.5 ‰ with an average $\delta^{13}C$ value of -33.5 ± 1.8 ‰ (n=10).

### 3.3 Organic carbon content and carbon isotope ratios of grassland soil and vegetation

The SOC content of the grassland soil in the upper layers is much lower than that in the forest soil and ranges between 0.4 and 1.5 % for GLA and between 0.7 and 1.7 % for GLP (0–5 cm, Figure 2c and 2d). Most grassland profiles show a similar relation between SOC content and depth: the SOC content decreases from values between ca. 0.7–1.3 % at the top to values of ca. 0.2 % below 1 m depth. However, there are exceptions: in the first few decimeters, the GLA-M and GLA-LM profiles have lower SOC than the other profiles. The valley profiles are also different: both have much higher SOC values than the other hillslope profiles in the top 30 cm: at the GLP profile, the SOC content of the valley profile is higher than that of other profiles down to a depth of ca. 100 cm. %OC do not differ significantly in the upper 0–50 cm depth for GLA and GLP for all profile positions (Supplementary Table 1).

On both grassland hillslopes, $\delta^{13}$C values are clearly higher for the valley profiles in comparison to the hillslope profiles, with the absolute differences being largest down to a depth of 40 cm for the GLA site and down to ca. 60 cm at the GLP site (Figure 3c and 3d). The variation of $\delta^{13}$C values with depth is similar on most hillslope positions under grassland: on all sites, $\delta^{13}$C values vary within a relatively narrow range (from -24 to -20 ‰) below a depth of 40 cm. Above 40 cm, some profiles (GLP-B, GLP-UM, GLA-UM and GLA-T) show a gradual decline of $\delta^{13}$C values with depth, with maximum $\delta^{13}$C values of ca. -16 ‰ at the top. In most profiles, however, the decline of $\delta^{13}$C with depth is much smaller or even non-existent. $\delta^{13}$C values of profiles under grassland (GLA and GLP) did not differ significantly in the upper 0–50 cm of the profiles (Supplementary Table 1). Soil $\delta^{13}$C values are consistently lower than those observed in the vegetation: the $\delta^{13}$C of the grassland vegetation ranges between -13.4 and -11.4 ‰ with an average value of -12.8 ± 0.6 ‰ (n=7).

### 3.4 Soil organic carbon stocks for grassland and forest soil profiles

For all hillslope positions, the absolute values of the cumulative SOC stocks were consistently higher in forest profiles, irrespective of the lower depth considered to integrate the SOC stock (Figure 4). However, in the valley profile, the absolute cumulative SOC stock under grassland is similar to that under forest for every soil layer. Both for the forested (F1 and F2) and grassland transects (GLA and GLP), no major differences in SOC stocks were observed between both profiles (Supplementary materials Table 1). SOC stocks at the valley position were significantly different from all other hillslope positions under grassland (Supplementary Table 2).

### 3.5 $^{14}$C analyses of bulk SOC

We measured the $^{14}$C activity of bulk SOC at different depths in both grassland and forest profiles (5–10 cm, 20–25 cm, 50–60 cm and 100–110 cm). The $^{14}$C activity of the bulk SOC is plotted against the $\delta^{13}$C of SOC in Figure 5. Along the forest transect, all surface samples at the top and valley positions showed a pMC value greater than 100 %, indicating a significant contribution from bomb-derived carbon (post-1950). The pMC values of the forest soil samples (F2-T and F2-V) decrease to 42–52 % at a depth of 50 cm. This decrease of pMC with depth is associated with a modest increase in $\delta^{13}$C.

For the grassland transect, samples from the soil surface layer (5–10 cm) on the hillslopes show clearly lower pMC values in comparison to the forest sites, ranging between 78 and 85 %. At the valley position, topsoil samples from both grassland transects have pMC values greater than 100 % (106 % and 103 % respectively), slightly higher than the pMC values observed for topsoils in the valley positions under forests. The pMC values of bulk SOC decrease with increasing depth down to 60 % at 100 cm depth (GLP-T, Supplementary Figure S6). The pMC values under grassland at depth are generally higher than the pMC values observed under forest at similar positions. Under grasslands, the decrease of pMC with depth is associated with a decrease of $\delta^{13}$C (Figure 5).

## 4. Discussion

### 4.1 Difference in carbon sources between grassland and forest soils

$\delta^{13}C$ depth profiles of forest soils (Figure 3a and 3b) are similar to those observed in the primary vegetated forest of Didy, located ca. 60 km south of our forest sampling locations (Winowiecki et al., 2017). They reported a median value of -27 ‰ in the forest topsoil and also found that $\delta^{13}C$ increased with depth. One partial explanation for the $^{13}C$ enrichment with depth under forest is the Suess effect which leads to a gradual decrease of the $\delta^{13}C$ values of atmospheric $CO_2$ over time (Keeling et al., 2017). In our forest profiles, an increase in $\delta^{13}C$ values of 2–3 ‰ over the upper 60 cm is observed (Figure 3a and 3b). Given the magnitude of the Suess effect (2 ‰ over the past 250 years; Keeling et al., 2017), the variations in $\delta^{13}C$ that we observe in the forest soil profiles may, to a large extent, be ascribed to the Suess effect. An additional explanation for $^{13}C$ enrichment with depth is commonly observed in soil profiles is that the soil microbial biomass is generally more $^{13}C$-enriched than its substrate, and soil organic matter therefore typically becomes more enriched in $^{13}C$ due to the incorporation of microbial biomass or microbially derived C (Ehleringer et al., 2000). It has been shown that SOC in deeper soil layers is in a more advanced stage of decomposition, and is increasingly composed of stable microbially derived SOC (Domeignoz-Horta et al., 2021; Kallenbach et al., 2016; Schlesinger, 1977).

The $\delta^{13}C$ values of forest vegetation (-32.5 ± 1.71 ‰) are consistent with a C3 vegetation signature. However, they are ca. 3 ‰ lower than the value estimated from the empirical relationship of Kohn (2010) that predicts the $\delta^{13}C$ values of C3 vegetation based on annual precipitation, latitude and altitude (-29 ‰). As the vegetation samples were collected by hand and thus taken not far above the soil surface, the lower measured values could be due to the "understory effect". The understory effect or canopy effect found in tropical forest is mainly related to the gradient in $\delta^{13}C$ of ambient $CO_2$ along the vertical gradient: $\delta^{13}C$-$CO_2$ is low close to the ground, due to elevated $CO_2$ concentrations via the contribution of soil respiration. Higher up in the canopy, the $\delta^{13}C$-$CO_2$ values are closer to the average atmospheric $CO_2$ composition. Kohn (2010) recommends -31.5 ‰ as a cut off value for identifying the understory effect in a closed-canopy forest such as the Zahamena National forest.

In contrast to the forest profiles, SOC profiles on the grassland transects show a decline of $\delta^{13}C$ with depth (Figure 3c and 3d). Some of our profiles, such as GLP-UM and GLP-B, show a very clear decrease of $\delta^{13}C$ in the upper 40 cm of the soil profile (from ca. -16 ‰ to ca. -22 ‰, Figure 3d). In most profiles, however, $\delta^{13}C$ values of -20 ‰ down to -23 ‰ are observed in the top layer and values decrease very gradually over the whole depth of the profile with minimum values of -23 ‰ to -24 ‰ at 2 m depth.

The $\delta^{13}C$ soil value of the SOC depends on the $\delta^{13}C$ value of the aboveground vegetation and the corresponding litter that gets incorporated into the soil profile over the period which is represented by the remaining SOC (Burney, 2004) (Bird et al., 2003). The $\delta^{13}C$ average values of the vegetation in the grassland transects (-12.8 ± 0.6 ‰) is a typical value for grassland plant species (C4 plants), and is consistent with the value found in a C4 species-dominated basin in Madagascar (-12.2 ± 0.7 ‰, Marwick et al., 2014). If C4 vegetation would have been dominant for a long time on the hillslopes, we would therefore expect topsoil $\delta^{13}C$ values typically between -15 to -13 ‰, with little variation with depth as observed in the few studies documenting

soil $\delta^{13}$C profiles under long-term C4 vegetation (Conrad et al., 2017). In contrast, while a few of our grassland soil $\delta^{13}$C profiles have topsoil $\delta^{13}$C values that are consistent with the current C4 vegetation (GLP-UM, GLP-B and GLA-T), most

350 surface soil samples show a $\delta^{13}$C value that is lower than would be expected of soils under C4 vegetation. This suggests that, even in the topsoil, an important fraction of the SOC is derived from C3 vegetation. Using the average $\delta^{13}$C of the forest soils as the C3 end member ($\delta^{13}$C = -25.3 ± 0.9 ‰) and using the values found by Conrad et al. (2017) as C4 end member (values range from -15 and -13‰ with mean value of -13.8 ‰), the contribution of C3 plant material to the SOC present in the upper 0–55 cm of these grassland soil profiles is estimated at ca. 70 %, with the exception of the valley position (Figure 6).

In contrast to the other grassland hillslope profiles, surface $\delta^{13}$C values of the valley positions are closer to the expected $\delta^{13}$C value of grassland vegetation, with an average value of -15.3 ± 0.8 ‰. For GLP-V, the $\delta^{13}$C values increase between the surface and 50 cm. This suggests a much higher incorporation of OC from C4 vegetation in the valleys in comparison to the hillslopes (Figure 6).

The same processes that cause an increase in $\delta^{13}$C with depth under C3 vegetation (Suess effect and microbial processing)
would be expected to also lead to an increase in $\delta^{13}$C with depth for grassland profiles if the latter would have fully developed under C4 vegetation. However, we observed an opposite pattern: $\delta^{13}$C decreases with depth under grassland vegetation (Figure 3c and 3d). Desjardins et al. (2013) observed a similar depletion in $\delta^{13}$C with depth in soils under tropical savannah (C4) vegetation in central Cameroon. At the surface, the $\delta^{13}$C values ranged from -16.4 to -13.9 ‰ and decreased towards -21.9 ‰ and -22.3 ‰ between 2 and 4 m depth. They interpreted this decrease in $\delta^{13}$C with depth as the result of a C3 to C4 vegetation
shift. While the relatively young SOC at the top of the profiles was primarily derived from C4 plants (resulting in a high $\delta^{13}$C), the older SOC in the lower part of the profile was primarily a relic from the time where the landscape was covered by forest (with a low $\delta^{13}$C). Sanaiotti et al. (2002) reported similar results from a savannah landscape in the Amazon region where the $\delta^{13}$C signatures of soil organic matter under savannah areas converged to the signature of a nearby forest at greater depths. Therefore, it was assumed that the savannah areas had replaced forested areas during the Holocene period (Sanaiotti et al.,
2002). Importantly, a decline of $\delta^{13}$C with depth towards the values reported in these studies and similar to those we observed is not always observed under C4 vegetation. For instance, Martin et al. (1990) describe profiles under C4 savannah where $\delta^{13}$C values are always above -17 ‰ down to depths of 120 cm and showed little systematic variation with depth, except for somewhat lower values for the deepest horizon that was sampled. This is to be expected in systems which have been under C4 vegetation for very long time spans: indeed, both the Suess effect and fractionation during microbial decomposition of SOC
are expected to lead to an increase in $\delta^{13}$C under C4 vegetation as well.

Similar to Desjardins et al. (2013) and Sanaiotti et al. (2002), we observed a clear convergence of the grassland soil profiles towards the forest signature of -24.0 to -23.0 ‰ with depth, except for the valley sites (Supplementary figure S7). At the surface, the average difference is ~6.6 ‰, declining to ~1.5 ‰ at 1 m depth. The average difference never becomes zero, except for the UM slope position suggesting that there may be some contribution from C4 plants to the SOC pool at all depths.
This pattern, where the $\delta^{13}$C value of grassland converges towards the $\delta^{13}$C value of forested soil with depth, is best explained by assuming that the vegetation of the grassland hillslopes we sampled has changed rather dramatically from a vegetation that

was most likely dominated by C3 plants (thus most likely a forest or wooded savannah) to an open vegetation completely dominated by C4 vegetation. The higher $\delta^{13}$C values we observed in the upper part of some of the profiles (GLA-T, GLP-UM and GLP-B) suggest that deforestation may have occurred in patches and that the locations of these profiles were deforested

first, resulting in topsoil SOC pools with a higher $\delta^{13}$C value in comparison to more recently deforested locations. The fact that the $\delta^{13}$C profiles measured at GLA-T, GLP-UM and GLP-B also converge to the forest profiles at depth suggests that also these locations originally had a tree-rich vegetation cover, but that this tree cover was removed earlier than on the other locations.

The results for the valley locations are different: here the SOC has a $\delta^{13}$C value reflecting a C4 vegetation down to 50 cm, and

also below this depth, the forest and grassland profiles remain significantly different, with higher $\delta^{13}$C values under grassland (p-value <0.001). The difference at depth compared to other locations suggests that the vegetation in the valleys has had an important fraction of C4 plants for a very long time allowing the slow cycling SOC pools stored at greater depth to acquire a $\delta^{13}$C signature significantly affected by C4 vegetation (ca. 43 %). In addition, we observed more dense grass vegetation in the valley positions compared to other profile positions (Supplementary figures S2 and S4). The higher $\delta^{13}$C values observed in

the upper part of the valley profiles (0–40 cm for GLP-V) suggest that also in these landscape positions the relative importance of C4 vegetation has recently increased, but lateral transfers may also play a role via erosion of recent C4 inputs from the upper slopes and subsequent deposition in the valley.

Within our profiles, we only observed pMC values > 100 % in some upper layers (<10 cm) of forest sites and at the valley position of the grassland soil transects (Figure 5). In contrast, the SOC at 5–10 cm depth in the grassland hillslope sites always

showed pMC values lower than 85 %. At the same time, the $\delta^{13}$C data indicate that a significant fraction (ca.~65 %) of the SOC at these sites is still C3-derived in most grassland profiles (Figure 6), despite the current dominance of C4 grasses. Thus, the $^{14}$C data corroborate the $\delta^{13}$C results and suggest that a significant fraction of the grassland soil SOC is relatively old and was formed when these locations were still under forest. This, in combination with a low degree of SOC replacement after the vegetation transition, results in the presence of a mixed $\delta^{13}$C signal in combination with a relatively low pMC. The low

replenishment/replacement of SOC from the grassland vegetation may be partly explained by (over)grazing and fire activity, both of which should reduce the organic inputs from C4 vegetation after deforestation (Abril et al., 2005; Bond and Keeley, 2005). A second mechanism possibly contributing to the lower $\delta^{13}$C and pMC values of surface soils along the grassland transect is that the soil might have been subject to more intense surface erosion after the vegetation shift, thereby constantly removing young SOC (with a higher $\delta^{13}$C value) towards the valley position. This is confirmed by soil erosion rates derived

from in situ $^{10}$Be analysis of the topsoil samples (5–15 cm) which indicates that on a convex hillslope, both under grassland and forest, erosion rates increase from the top towards the valley position, where the erosion rates are consistently higher under grassland when compared to forest (L. Brosens et al., unpublished data).

For the grassland transects, significant differences were observed in %OC, $\delta^{13}$C and SOC stocks between the valley and other hillslope positions (T, UM, M, LM and B, Supplementary Table 2). SOC stocks and surface $\delta^{13}$C values on the upper slope

were lower than at the valley positions. $\delta^{13}$C values on the upper slope positions (T, UM, M, LM and B) were slightly higher

than those of forest soils at corresponding depths (Supplementary figure S7). Erosion rates in the upslope positions are expected to have increased after deforestation. Restrepo et al. (2015) have reported a 33 % increase in erosion rates 30 years after deforestation in the Colombian Andes. This erosion leads to losses of SOC (Don et al., 2011; Rabetokotany-Rarivoson et al., 2015). Even though surface erosion itself is not associated with isotope fractionation, it indirectly influences SOC $\delta^{13}$C values by removing the surface layer and exposing the deeper soil layer, where SOC is a mixture of SOC from the subsoil and fresh OC from the vegetation. Thus, erosion can lead to different $\delta^{13}$C values (Häring et al., 2013), and this process could contribute to the higher SOC stocks and $\delta^{13}$C values observed in the valley positions for the grassland transect. However, under forest (C3 vegetation), no substantial differences in the $\delta^{13}$C are observed along the transects (Supplementary Table 2). This suggests that erosion might not play a major role in the variation of $\delta^{13}$C in a forest dominated landscape as significant erosion would have led to SOC accumulation in the valley position.

The high pMC of SOC in the grassland valley position (GLP-V) is consistent with the idea that this site acts as a depositional area for recently eroded C. It can be noticed that also in the valley sites, pMC decreases with depth (Supplementary Figure S6), which is consistent with a significant local production of SOC. On the other hand, pMC values in the valley sites are clearly higher than those observed on the forest and grassland hillslopes, at least for depths < 55 cm: this suggests a significant addition of young eroded SOC. Surface erosion is expected to be variable across topographic positions along the hillslope transect, with minimal impact at the top and maximum impact on the steeper sections towards the bottom of the hillslope transect. However, no consistent variation of topsoil pMC values was observed along the grassland transect (Supplementary Figure S6). Within the grassland profiles, there is a clear positive relationship between $\delta^{13}$C values and $^{14}$C activity (R²=0.7, P-value = 0.006, Figure 5), whereby older SOC corresponds to a higher contribution of C3-derived (forest) C. In contrast, for sites under current forest vegetation, the upper soil layer contains mostly young OC (pMC >100 %), and these profiles are dominated by C3-derived C throughout the range of $^{14}$C activities they represent. In addition, pMC values of grassland soil profiles converge to the pMC values observed in the forest soil profiles at greater depths (Figure 5).

## 4.2 Response of SOC stocks to vegetation transition

The average SOC stock we measured in the upper layer (0–30 cm) of our grassland transect of 33 ± 9 Mg C ha$^{-1}$ and 30 ± 11 Mg C ha$^{-1}$ (Supplementary Table 3) is clearly lower than the average stock found in ferralsols under grassland in Madagascar, with an average of 77 ± 33 (n=8) Mg C ha$^{-1}$ and is in the same range as Luvisols and Arenosols under grassland with an average of 49 ± 33 Mg C ha$^{-1}$ (n=4) (Chevallier et al., 2020).

The SOC stocks in our forest transect of 70–74 Mg C ha$^{-1}$ (0–30 cm) and 137–139 Mg C ha$^{-1}$ (0–100 cm) are similar to that of Grinand et al. (2017) who conducted a field study in the southeast humid forest of Madagascar and measured SOC stocks of 70–131 Mg C ha$^{-1}$ in the upper 30 cm and 139–296 Mg C ha$^{-1}$ in the upper 100 cm. In addition to this, they found that the SOC stock in the upper 30 cm accounts for half (49 %) of the SOC stored in the upper 100 cm. Rabetokotany-Rarivoson et al. (2015) observed a wide range of SOC stocks between 39–233 Mg C ha$^{-1}$ for the upper 30 cm in the humid eastern forest. The average values they reported for both soil layers were 110 Mg C ha$^{-1}$ (0–30 cm) and 226 Mg C ha$^{-1}$ (0–100 cm), showing that 49 % of

the SOC of the 0–100 cm soil layer is stored in the upper 30 cm. Andriamananjara et al. (2017) measured SOC stocks sampled
in the eastern humid forest of Madagascar and reported an average cumulative SOC stock of 90 Mg C ha$^{-1}$ (0–30 cm) and 137
Mg C ha$^{-1}$ (0–100 cm), which are also similar to the reported values in our study.

By comparing the absolute value of cumulative SOC stocks under forest and grassland, it can be seen that total stocks are
clearly higher under forest, except in the valley (Figure 4).

The first-order control on the SOC stock of the soil is the balance between inputs and outputs of carbon in soils (Davidson and
Janssens, 2006). These inputs include vegetation such as litter and roots from biomass, while the outputs include $CO_2$
emissions, erosion and leaching of dissolved organic matter into the groundwater. Deforestation has a significant impact on
SOC stocks, primarily because carbon inputs are reduced. Grinand et al. (2017) estimated an average change in SOC stocks of
-10.7 % and -5.2 % in the 0–30 cm and 0–100 cm soil layers when forest is converted to grassland in the humid ecoregion of
Madagascar over a 20 years timespan. In addition, a study of tropical soils by Don et al. (2011) reports an estimated -12 %
change in SOC stock when the primary forest is converted to grassland over 25 years timespan. However, when comparing
the grassland SOC stocks with the forest SOC stocks for our profiles, an average loss of -55.7 % is observed in the upper 30
cm soil layer, which is much higher than in the aforementioned studies. Several factors may explain this difference. The studies
of Don et al. (2011) and Grinand et al. (2017) report changes over a period of 20–25 years after deforestation. The fact that we
find a much larger reduction in SOC stock may be due to the fact that deforestation has occurred much longer ago at the sites
we sampled. It is often assumed that the major change in SOC occurs during the first five years after deforestation and a new
SOC equilibrium is reached between 20–40 years (Cerri et al., 2007). Our data suggest that this may not be the case in the
environment we studied: this may be due to the large difference in SOC production between grasslands and forests. Indeed,
our data suggest that even today, a large fraction of the SOC that is present under grassland was produced when the grassland
sites were under C3 vegetation. Similar observations were reported by several authors in landscapes where forest vegetation
has been transformed into grassland more than thousand years ago, showing evidence that C3-derived carbon can remain
present for millennia after the replacement of the original vegetation (Desjardins et al., 2013; Guillet et al., 2001; Schwartz et
al., 1986). Rabetokotany-Rarivoson et al. (2015) and Razafindrakoto et al. (2018) found a strong difference in SOC stocks
between the initial forest vegetation and the final stage of deforestation (i.e. a landscape dominated by grasses). This suggests
that due to the low SOC productivity of savannah systems, the time needed to reach a new equilibrium SOC stock is indeed in
the order of centuries to millennia rather than decades (Wutzler and Reichstein, 2007).

As opposed to what we found on the other hillslope positions, the SOC stocks under forest and grassland are relatively similar
in the valley positions (Figure 4). There are two mutually, non-exclusive, mechanisms that can explain this observation: first,
the valleys may have been permanently covered by C4 vegetation that is much more productive than the vegetation on the
hillslopes. This is indeed highly likely: the valleys are clearly much wetter than the hillslopes and are locations where nutrients
eroded from the hillslopes may accumulate. Second, part of the SOC found at the valley locations may not have been produced
locally but may result from lateral transport of SOC through erosion from the hillslopes.

## 5. Conclusions

We report soil organic carbon and $\delta^{13}$C profiles from hillslope transects in both forested and grassland sites in the Lake Alaotra region, central highlands of Madagascar, a region characterized by a high density of intense erosion features (lavaka). The $\delta^{13}$C values of the forest soil profiles are typical of soils developed under C3 vegetation for a very long time. In contrast, the $\delta^{13}$C values of the grassland soil profiles decreased with depth and showed a clear convergence towards the forest $\delta^{13}$C signature at depths below ~80 cm. These $\delta^{13}$C trends indicate a C3 to C4 vegetation shift in the present-day grasslands close to the Zahamena National forest. While the SOC stocks of the forest were in the same range as those reported in other parts of Madagascar's humid forest ecoregion, grassland SOC stocks were substantially lower (by ~48–56 % over the entire 2 m profile) than those of the forest. Topsoil (0–5 cm) $\delta^{13}$C values under grassland indicate that the middle and the lower-middle profiles were most vulnerable to erosion and that the valley position acts as a depositional zone. Despite the absence of substantial new inputs from C3 vegetation, the bulk of the SOC stocks remains largely dominated (70 %) by (old) C3-vegetation. $^{14}$C measurements of bulk SOC of the topsoil in the top, middle and lower-middle positions under grassland furthermore show that soil organic carbon contains less modern carbon at the surface. For the valley position of our grassland transects, the surface SOC with a higher $\delta^{13}$C consists of modern carbon indicating a late expansion of C4 vegetation. Overall, in our study area where human pressure is high, current grass covers have expanded strongly at the detriment of forest. The hillslope soil profile data give explicit information on the aboveground vegetation shift in the past. Our study does not only show that these changes are indeed reflected in the OC characteristics in the hillslope and valley soils of the area, but also allows to make an estimate of the impact of anthropogenic deforestation and water erosion on soil carbon inventories in a tropical environment. A strong decline of the soil organic carbon content is detected under grassland, which is only half of that under forest conditions. This very strong reduction can be attributed to the low net C4 input since deforestation: more than half of the OC stored under grasslands is still C3-derived. In addition, there is a limited contribution of C4 grass production to the soil organic carbon pool. This indicates that the time since deforestation is likely to be reflected in the fraction of the SOC pool that has been mineralized or lost. Our results are consistent with the hypothesis that a vegetation shift has occurred in the Lake Alaotra region, and offer a promising avenue to expand this approach on a wider scale to help understand the vegetation cover changes in the central highlands of Madagascar. A soil organic carbon model combining stable and radiocarbon isotope data could offer a valuable tool to determine the timing of the vegetation change in the Lake Alaotra landscape.

**Data availability**

All data generated in this study are available in the Supplementary file.

## Author contributions

G.G, S.B., B.C., L.J with contribution of L.B and V.F.R. designed the study project. T.R. and T.R. and L.J. helped supervise the project. L.B. and V.F.R. planned fieldwork and collected samples. M.D. contributed to forest profile sampling and the analyses of these samples. V.F.R. analyzed grassland soil samples and wrote the manuscript with S.B and M.D. All authors provided critical feedback and helped shape the research, analysis and manuscript.

## Competing interests

The authors declare that they have no conflicts of interest.

## Acknowledgements

This research is part of the MaLESA (**Ma**lagasy Lavaka, **E**nvironmental reconstruction and **S**ediment **A**rchives) project funded by KU Leuven (Special Research Fund). Travel and research grants were provided by YouReCa and FWO (11B6921N, 12Z6518N,V436719N). We thank two anonymous reviewers for their constructive inputs, which greatly improved our manuscript. We thank Z. Kelemen, L. Fondu, C. Coeck, C. Morana, E. Vassilieva, J. Verdonck and Y. Stroobandt (KU Leuven), M.P. Razafimanantsoa, M. Rakontondramanana, M.E. Rakotonirina and F. Raharison (LRI, Antananarivo), and M. Boudin (KIK/IRPA, Brussels) for technical and analytical assistance. We also thank M. Van de Broek, N.R.G. Voarintsoa, R. Cox and Pr. M. Mietton for constructive input and discussions. We thank MNP (Madagascar National Parks) and MEDD (Ministère de l'Environment et de Development Durable) of Madagascar for authorizing sample collection in the protected area of the Zahamena National Park.

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

710

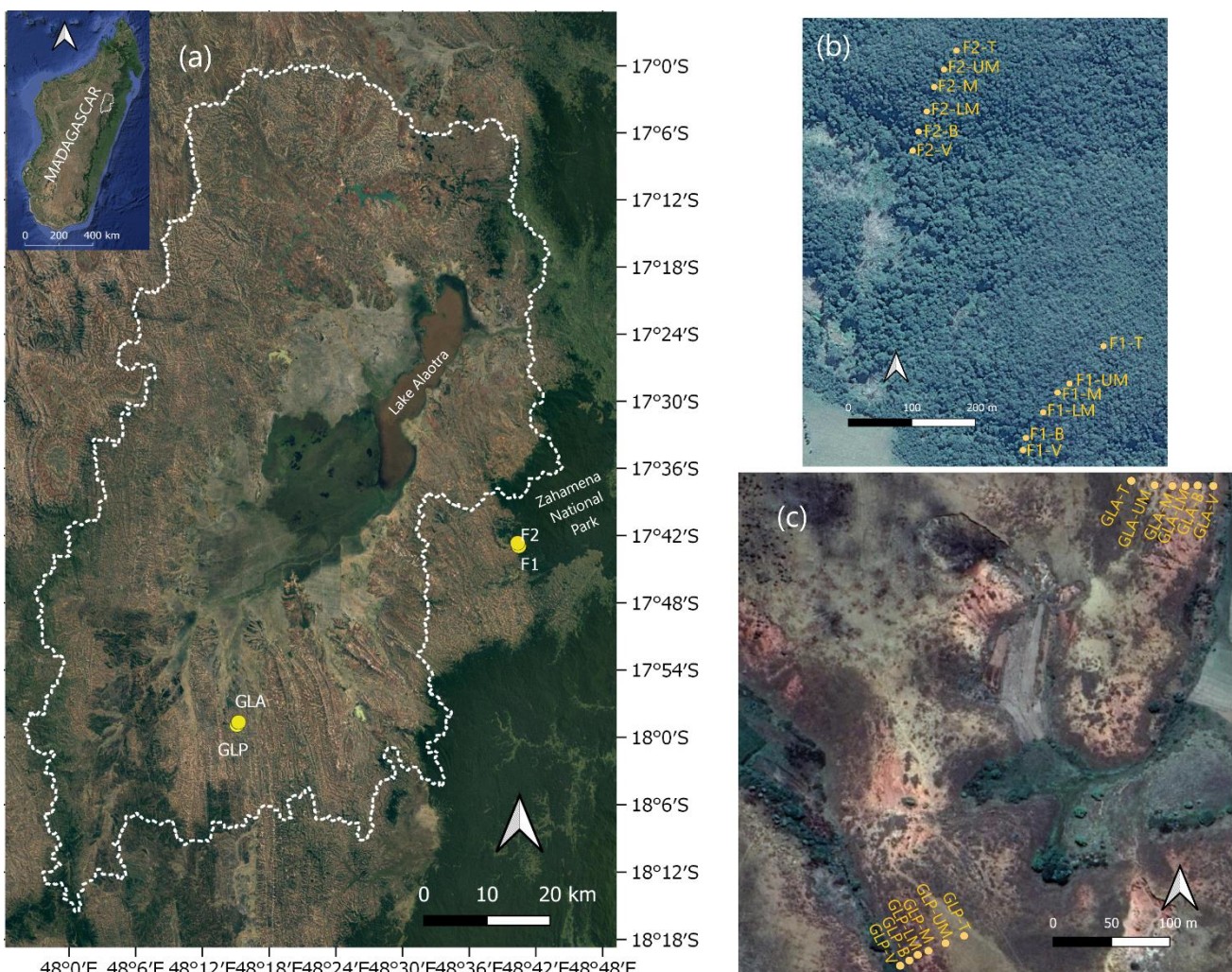

**Figure 1: (a) Location of the study region with indication of the sampled hillslope profiles. The dotted white line delineates the watershed of the Lake Alaotra Basin. GLA and GLP are the two convex hillslopes under grassland, F1 and F2 the two forested convex hillslopes in the Zahamena National Park. Satellite view of the two forest transects (b) and the two grassland transects (c). (© Google Earth 2020).**

715

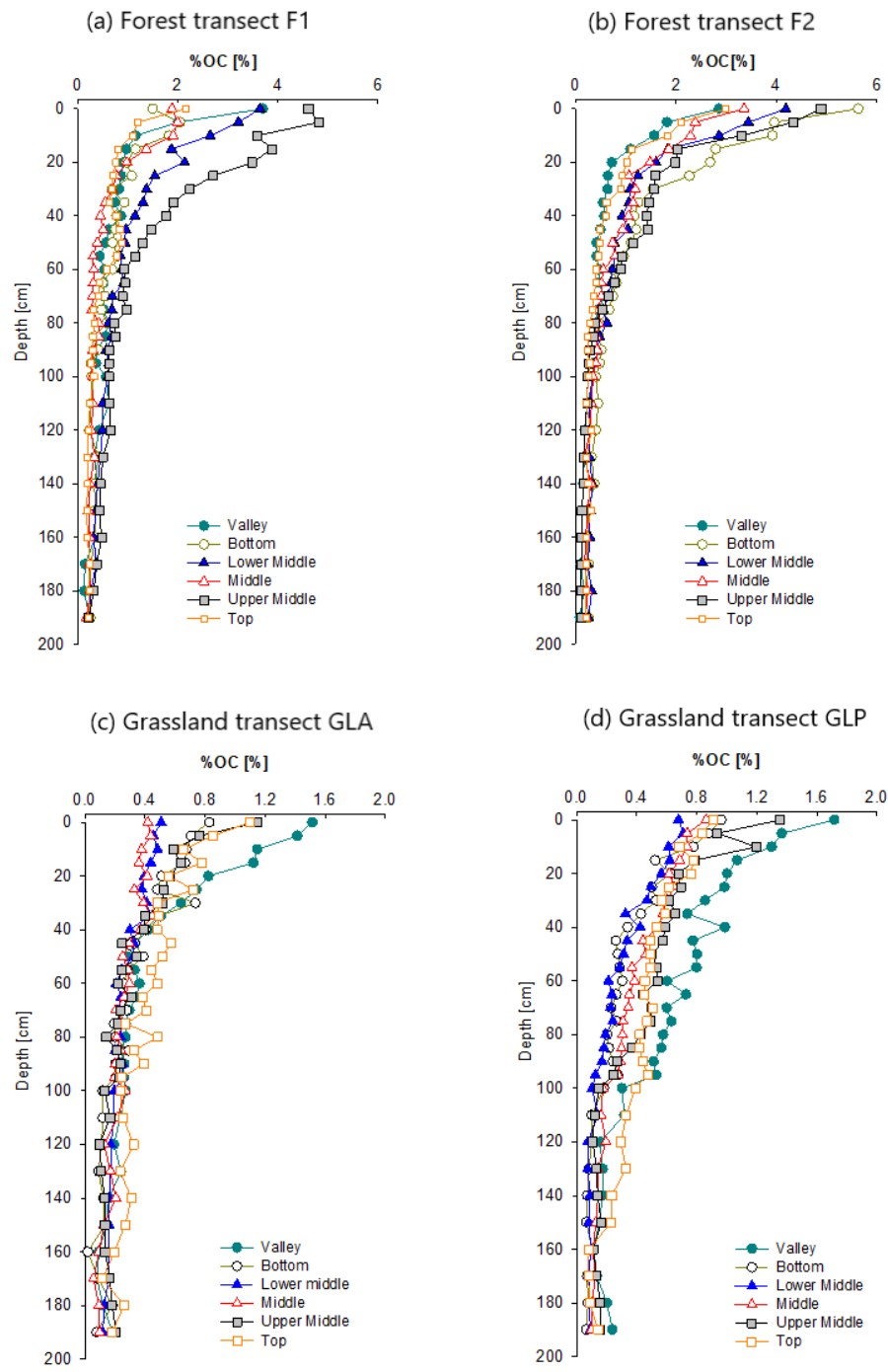

**Figure 2: Depth profiles of %OC for each sampling location (Top-Upper Middle-Middle-Lower Middle-Bottom-Valley) of the four transects including F1 (a), F2 (b), GLA (c) and GLP (d).**

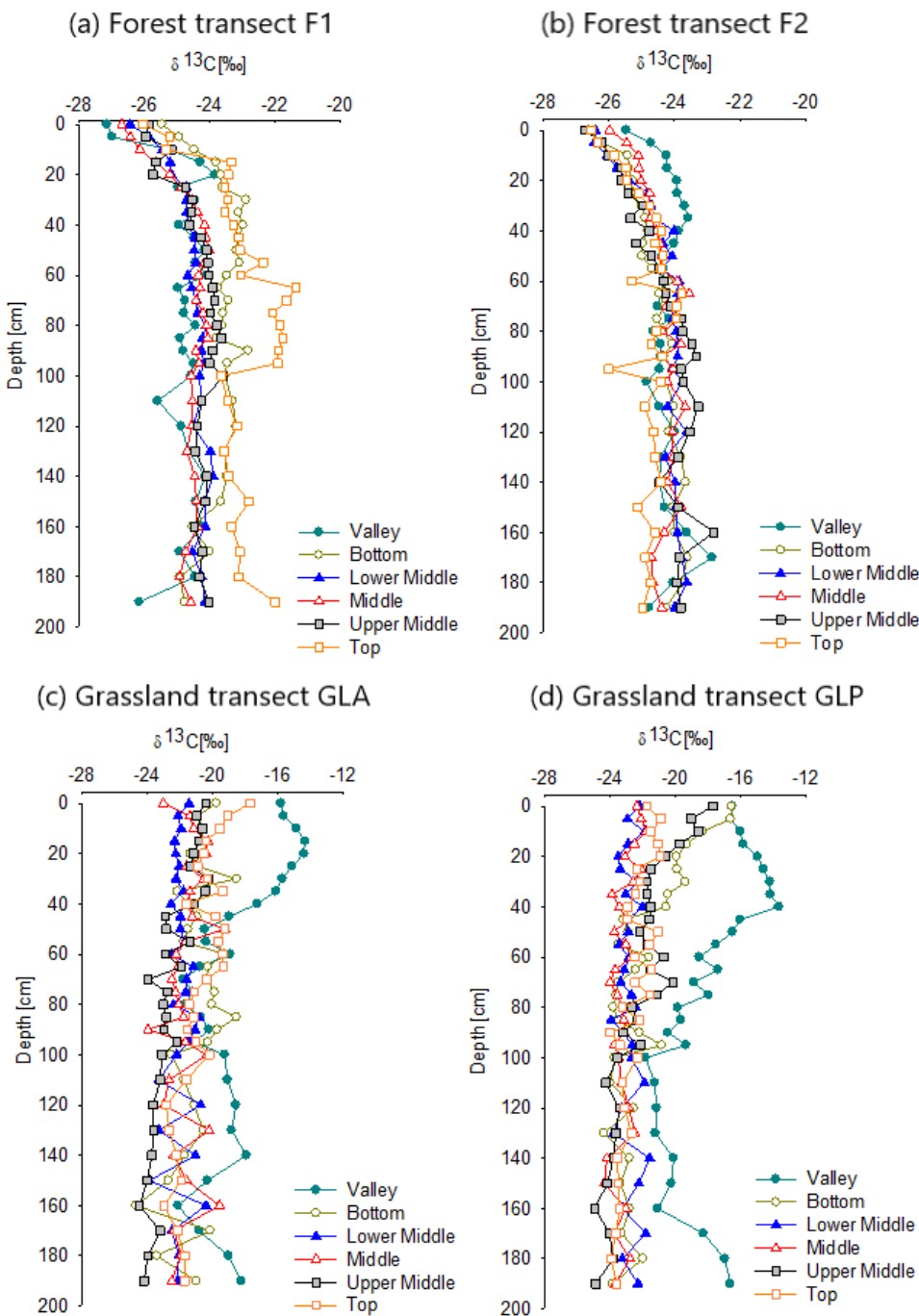

**Figure 3: Depth profiles of δ¹³C for each sampling location (Top-Upper Middle-Middle-Lower Middle-Bottom-Valley) of the four transects including F1 (a), F2 (b), GLA (c) and GLP (d).**

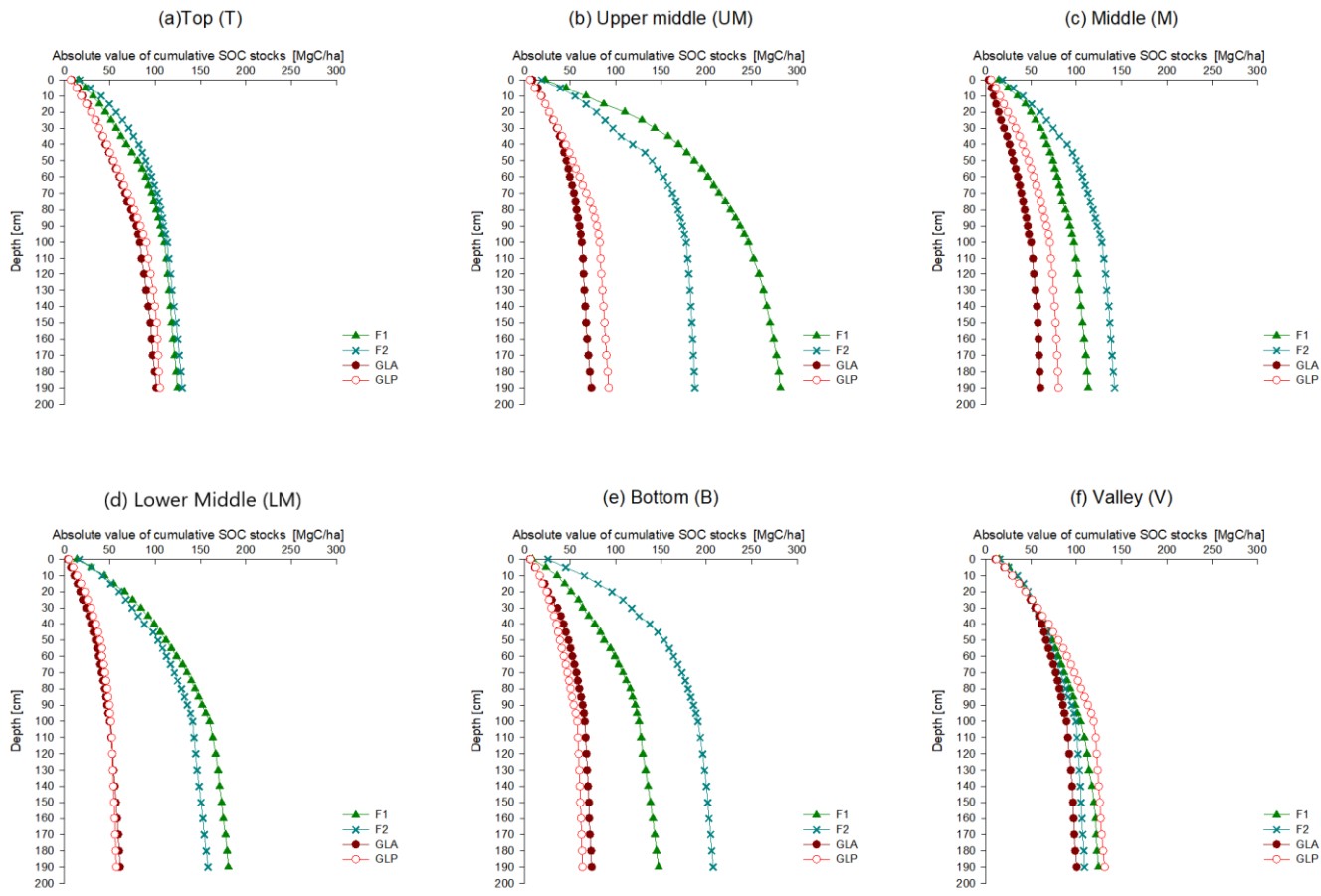

**Figure 4: Absolute values of cumulative SOC stocks of hillslopes F1, F2, GLA and GLP plotted together for each sampling location: (a) Top, (b) Upper Middle, (c) Middle, (d) Lower Middle, (e) Bottom and (f) Valley.**

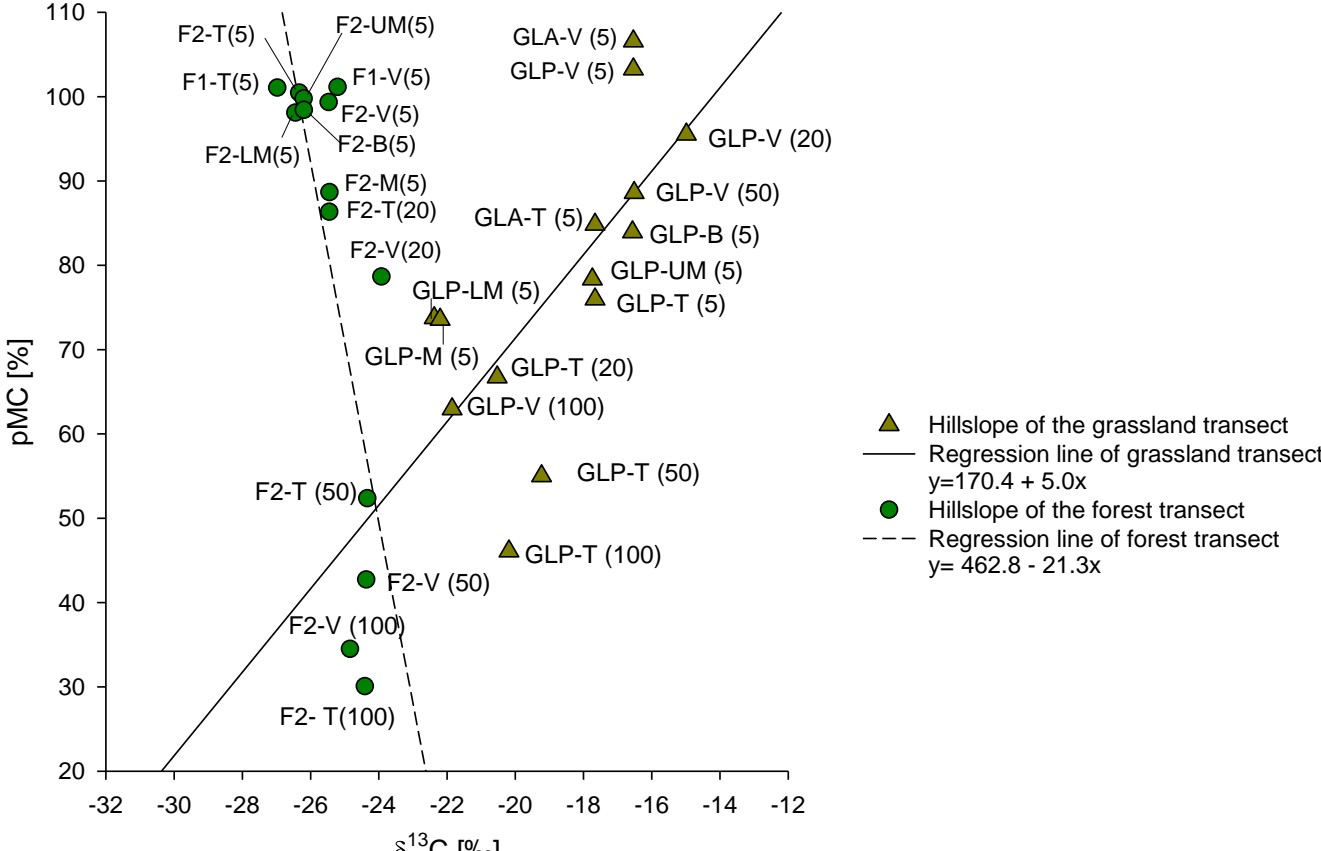

**Figure 5: Comparison of the average value of ¹⁴C-pMC of bulk SOC plotted against δ¹³C of SOC of grassland and forested soil. Each point is labelled by the transect position and the top depth interval. Continuous and dotted black lines represent the regression lines for grassland soil data and forest soil data, respectively.**

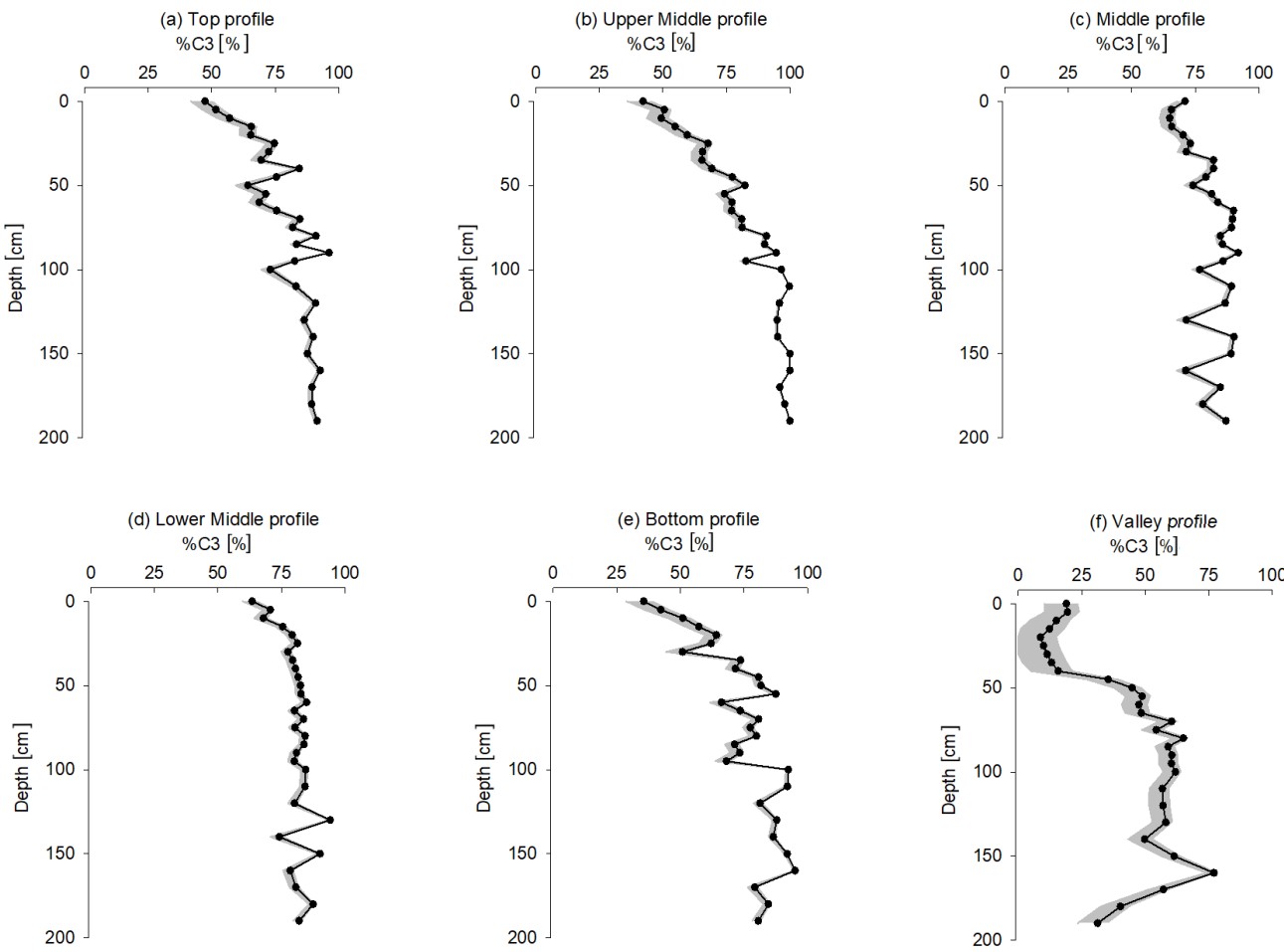

730

**Figure 6:** **Variation with depth of the fraction of remaining forest SOC for the grassland profiles. The black line represents the variation of %C3 by taking -13.8 ‰ as the reference value of δ¹³C SOC under permanent C4 vegetation. The grey area represents the range of %C3 by using a reference range of -15 ‰ to -13 ‰ as the δ¹³C SOC under permanent C4 vegetation.**

735