# Peer review of "Stable isotope profiles of soil organic carbon in forested and grassland landscapes in the Lake Alaotra basin (Madagascar): insights in past vegetation changes."

_Biogeosciences, 2021_

## Referee Comment (RC1)

*General comments*

I have read with interest this paper, which describes the consequences of vegetation change and erosion processes on SOC dynamic and stocks.

It is an interesting research objective, and the purposes of this work would fall within the aims of this journal. In general, I think the paper is interesting and has potential. However the manuscript needs some improvements, outlined in the specific comments, but its main shortcoming is outlined below.

The study is based on the comparison of toposequences under forest and grassland and the assumption that the soils under these different vegetations were identical or at least very similar before the vegetation change. However, the paper gives almost no information on these soils, either from a chemical or physical aspect. Some parameters, such as texture, have a strong link with the dynamics and stocks of organic matter. How can we be sure that the very large decreases in C stocks observed under pasture is indeed due to deforestation and the erosion it induces, if we do not know that the soils are really comparable? A presentation of the main characteristics of the soils (if only in the supplementary material) is necessary before we can put forward the hypotheses set out in the discussion.

This manuscript, after the necessary improvements and corrections, would be acceptable for publication.

*Specific comments*

Abstract

Lines 17-18: the time span allowed by the δ13C to study the past dynamic of soil carbon ranges from years to millennia (rather than centuries)

Line 20: the SOC is low, not extremely low.

Line 23: "…which show typical profiles under C3 vegetation, with a slight increase with depth."

Line 30-31: "…suggesting a recent expansion of grass vegetation, and/or that the valleys are depositional areas from organic matter eroded from the hillslopes."

Lines 31-33: "Our approach, based...determine changing vegetation cover". This is true, but it has already been done in different parts of the world and published in many publications in the last 40 years. As this sentence is written, it sounds like a new approach.

Introduction

Lines 87-90: "The stable carbon isotope ratio…show a different degree of isotope fractionation". It is necessary to cite references

Materials and methods

Line 101: the rainfall is not very high; many tropical regions have average annual rainfall between 1500 and 3000 mm or more.

Line 103: "the mean annual temperature varies between 18 and 24°C" Really? Not the mean monthly temperature?

Line 118-120 AND Figure S3: the length and the gradient of the hillslopes are different under forest and grassland. Could this have an effect on erosion processes?

Line 123-126: Why is there such a large distance (about 60 km) between the soil profiles under the forest and those under the grassland? Were there no adequate situations for the grassland soils closer to the forest?

Important information about the soils is missing, which could be in the supplementary material: are the soils under forest and under grassland really similar, in chemical and physical terms. One of the objectives of the paper is to assess the effect of vegetation change on carbon stocks. Several soil parameters, such as texture, can influence organic matter stocks, so it is important to know whether the soils are similar.

Results

Line 207-208: The description of the C profiles is too brief and even wrong! For example, for the F1UM profile the SOC content varies from 60 to 200 cm, between 0.3 and 0.9 %, not 0.1 and 0.2 %.

Line 210-211: The description of the $\delta$ 13C profiles is too brief.

Line 218-219: It would be better to say that in the first few decimeters, these two profiles have lower SOC values than the other profiles.

Line 236: The sentence "However, the cumulative…on the GLP hillslope" is unnecessary.

Figure 3c: THIS IS NOT THE GOOD ONE!

Discussion

Lines 253-277: All these explanations of the evolution of $\delta$ 13C values under C3 forest vegetation are excessively long. Since the end of the 80's, many articles have detailed this. This does not provide decisive information to answer the objectives of the paper.

Lines 293-295: I do not agree, in the topsoil (what depth exactly?), the C3 contribution is much lower than 70%! See the figure 6.

Lines 297: for GLP-V the $\delta$13C value **increases** between the surface and 50 cm.

Lines 356-358: repetition of the lines 348-350

Line 380: "…, while the outputs include CO2,…" or "…, while the outputs include CO2 **emissions**,…"?

Line 397-398: It is not true that all the studies cited found strong differences in SOC stocks between savannah and forest situations. Moreover, the stocks are not calculated and commented on.

Lines 400-401: That is true, but what does it add to the discussion, at this point. It would be better to delete this sentence.

Line 411: "The δ13C values of the forest profiles increased with depth, which is expected for soils developed for soils developed under C3 vegetation". It would be better to say that these 13C profiles are typical of soils under C3 vegetation for a very long time.

Lines 417-418: you cannot say that organic carbon input from the new grassland vegetation is not significant: it represents almost a **third** of the carbon stock!

Line 429: "This indicates that the response time to deforestation depends on the rate of depletion of the old C3 pool." What does this sentence mean?

***Technical corrections***

Introduction

Line 42: Voarintsoa et al., not Voarintsoa and Cox

Materials and methods

Figure 1a: in the caption, it is written "dotted black line", but it is a "dotted white line".

Line 121: The supplementary material S3 does not show vegetation

Line 148: in the equation, δ13C, not δ13.

Line 197: for D(i), the unit of measurement is missing.

Results

Figure 2: in the caption: "middle" not "middles"

Line 207: the topsoil samples are 0-5 cm not 0-10 cm

Line 209: "…between -25.5 and - 27.1‰ …"

Discussion

Line 225-226: verify the profiles which show gradual decline: GLP-B, GLP-UM, GLA-T (not GLA-M)

Line 240: "at different depths" appears two times

Line 280: "…values of -20 down…" The symbol ‰ is missing.

Line 350: In the references, Brosens et al. is indicated as published in 2022.

---

## Author Comment (AC1)

**Author response to referee #2**

We thank reviewer #2 for his/her constructive comments and suggestions to improve our manuscript. Below we have formulated a first reply to the main concerns raised by the reviewer, where we provide an overview of the main changes we intend to make in the revised version of our manuscript:

- We will add information regarding the physical characteristics of the soil under forest and grassland, where we show that both sampling locations have comparable initial conditions, providing more confidence to ascribing the observed differences in SOC and $\delta^{13}$C to changes in vegetation.
  - The topographical characteristics (hillslope length and slope gradient) of the forested and grassland transects will be carefully compared, where additional information on the slope gradient of all transects will be added. Slope gradients are very similar for all sampled transects, where the transects under forest are slightly longer than the grassland ones. The possible influence on erosion processes will be discussed in a revised version of the manuscript.
  - Soil texture data for all soils will be discussed, where no significant differences between grassland and forested hillslopes are observed.
- An improved description of the $\delta^{13}$C and SOC profiles for the grassland and forest soils
- Clarification of some of the statements made, where additional information or references will be added where needed.
- Inconsistencies in figure captions, references, equations and units will be resolved and verified throughout the manuscript.

Reviewer comments are indicated in *italics*, our responses in regular font.

Response to reviewer #2

*GENERAL COMMENTS*

*I have read with interest this paper, which describes the consequences of vegetation change and erosion processes on SOC dynamic and stocks. It is an interesting research objective, and the purposes of this work would fall within the aims of this journal. In general, I think the paper is interesting and has potential. However the manuscript needs some improvements, outlined in the specific comments, but its main shortcoming is outlined below. The study is based on the comparison of toposequences under forest and grassland and the assumption that the soils under these different vegetations were identical or at least very similar before the vegetation change. However, the paper gives almost no information on these soils, either from a chemical or physical aspect. Some parameters, such as texture, have a strong link with the dynamics and stocks of organic matter. How can we be sure that the very large decreases in C stocks observed under pasture is indeed due to deforestation and the erosion it induces, if we do not know that the soils are really comparable? A presentation of the main characteristics of the soils (if only in the supplementary material) is necessary before we can put forward the hypotheses set out in the discussion. This manuscript, after the necessary improvements and corrections, would be acceptable for publication.*

REPLY: We thank the reviewer for their overall positive evaluation and the detailed suggestions to improve the manuscript. To test our hypothesis whether the differences in SOC and $\delta^{13}C$ between grassland and forest profiles are linked to vegetation changes, we agree that additional information on our soil transects would be valuable. Therefore, additional information on the slope gradient of all transects will be provided in the revised manuscript. We found that slope gradients are similar for all transects, even though the lengths of grassland transects are slightly shorter than the forest ones. We will also include information on soil texture data, which are available and show no significant differences between soils under grasslands and forest. Other specific comments have been addressed point-by-point in our replies below.

*SPECIFIC COMMENTS*

*Abstract*

*Lines 17-18: the time span allowed by the $\delta^{13}C$ to study the past dynamic of soil carbon ranges from years to millennia (rather than centuries)*

REPLY: Thank you for this, we will modify "centuries" to millennia as suggested.

*Line 20: the SOC is low, not extremely low.*

REPLY: Thank you for this, "extremely low" will be changed to "low".

*Line 23: "…which show typical profiles under C3 vegetation, with a slight increase with depth."*

REPLY: Thank you for this suggestion, the sentence will be rephrased as suggested.

*Line 30-31: "…suggesting a recent expansion of grass vegetation, and/or that the valleys are depositional areas from organic matter eroded from the hillslopes."*

REPLY: Thank you for this suggestion, the sentence will be rephrased as suggested.

*Lines 31-33: "Our approach, based…determine changing vegetation cover". This is true, but it has already been done in different parts of the world and published in many publications in the last 40 years. As this sentence is written, it sounds like a new approach.*

REPLY: We agree that this approach has been previously applied in different parts of the word, however not yet in Madagascar. We have therefore rephrased this sentence as follows "The method we applied, which is based on the large difference in $\delta^{13}$C values between the two major photosynthetic pathways (C3 and C4) in (sub)tropical terrestrial environments, provides a relatively straightforward approach to quantitatively determine changing vegetation cover in Madagascar."

*Introduction*

*Lines 87-90: "The stable carbon isotope ratio…show a different degree of isotope fractionation". It is necessary to cite references*

REPLY: The following reference will be added: Cerling and Harris (1999).

*Materials and methods*

*Line 101: the rainfall is not very high; many tropical regions have average annual rainfall between 1500 and 3000 mm or more.*

REPLY: We agree - the term "high" will be removed.

*Line 103: "the mean annual temperature varies between 18 and 24°C" Really? Not the mean monthly temperature?*

REPLY: We thank the reviewer to notice this, we will correct this statement accordingly: "The mean annual temperature is 20.6°C, ranging between 11°C in July and 28°C in January (Ferry, 2009)".

*Line 118-120 AND Figure S3: the length and the gradient of the hillslopes are different under forest and grassland. Could this have an effect on erosion processes?*

REPLY: We agree that the sampled hillslope transects under forest are longer (217 and 184 m) than the sampled grassland profiles (62 and 70 m). However, the slope gradients (derived from the 12 m resolution TanDEM-X DEM) of the four transects are comparable, with maximum slope gradients of 30° and 25° for the forest transects and 29° and 25° under grassland. In the revised manuscript, the supplementary Figure S3 will be improved and a revised version of this Figure will include the change of the gradient along the transects, a provisional version is shown below.

[Figure]

The two main types of soil erosion on hillslopes are water erosion and diffusive erosion. Water erosion rates typically increase with increasing slope length and gradient (Govers et al., 1994). Diffusive erosion fluxes are approximately proportional to the slope gradient (Heimsath et al., 2005; Pelletier and Rasmussen, 2009; Roering et al., 1999).

Based on the topographical characteristics only (i.e., assuming the same vegetation cover) of our hillslopes, we can thus expect diffusive soil erosion fluxes to be similar for all four transects which would result in lower diffusive erosion rates on the forested slopes as they are longer.  Similarly, one might expect higher water erosion rates on the the lower half of the forest transects when considering topography only as these are longer than the grassland profiles. However, the effect of slope length on water erosion rates is non-existent under dense, natural vegetation (Cerdan et al., 2010, Zhao et al., 2021) and it is therefore unlikely that there would be significant differences in erosion rates between the grassland and forest slopes if they would only have a different topography. The differences in erosion rates due to differences in topography are more than likely far less important than those related to differences in vegetation cover. Water erosion rates are minimal under forest, given the protection provided by the dense vegetation cover (Cerdan et al., 2010; Zhao et al., 2021).  A grass cover that is well below 100% does offer far less protection: consequently, actual water erosion rates may be expected to be significantly higher on the grassland slopes in comparison to the forest slopes (Carroll et al., 2000; Silburn et al., 2011; Zhao et al., 2021).

*Line 123-126: Why is there such a large distance (about 60 km) between the soil profiles under the forest and those under the grassland? Were there no adequate situations for the grassland soils closer to the forest? Important information about the soils is missing, which could be in the supplementary material: are the soils under forest and under grassland really similar, in chemical and physical terms. One of the objectives of the paper is to assess the effect of vegetation change on carbon stocks. Several soil parameters, such as texture, can influence organic matter stocks, so it is important to know whether the soils are similar.*

REPLY: We agree with the reviewer that the distance between the forest and grassland profiles is relatively large. The main rationale behind the site selection was that (i) grasslands on the western side of Lake Alaotra were the main focus, as these represent a large and continuous/homogeneous area with characteristic vegetation cover, for which we hypothesized that vegetation changes (deforestation) may have occurred long enough in the past to result in differences in SOC inventories and characteristics. The nearest zone of pristine forest is located on the eastern side of Lake Alaotra – given the wide alluvial plain that results in a fairly high distance between sites. While grasslands area are also present on the eastern side of the lake, they represent a much more narrow strip of land which may have been deforested relatively recently so that SOC inventories might still reflect the forest cover that was present until

recently. However, we paid careful attention to ensure that the topography of the transects was as equivalent as possible.

We agree that the chemical and physical characteristics of our soil should be comparable in order to verify our hypothesis of a shift in vegetation. The soils at both the forested and grassland sampling site are defined as ferralsols (Andriamananjara et al., 2017). We further verified the assumption of comparable soils by analysing the texture of the soil under forest and grassland. These results will now be included, we did not observe significant differences in texture of soils under grassland and forest (p-value =0.663 (sand); p-value=0.723 (silt) and p-value= 0.232 (clay)). In the new version of the manuscript, we will add the soil texture diagram (see below) to the supplementary figures, add a paragraph describing the used method to derive the soil texture and report the results of the texture analysis in the text.

[Figure]

**New supplementary Figure**: Texture triangle (clay, silt and sand) of soil under forest and grassland.

*Results*

*Line 207-208: The description of the C profiles is too brief and even wrong! For example, for the F1UM profile the SOC content varies from 60 to 200 cm, between 0.3 and 0.9 %, not 0.1 and 0.2 %.*

REPLY: We apologise for the error. We have verified and corrected these numbers and have further elaborated the description of these results:

"The OC content (%OC) of the forest profiles ranged between 1.5 and 4.8% for F1 and between 2.9 and 5.6% for F2 in the upper 0-10 cm (Figure 2a and 2b and Table S1). Overall, the %OC trends of the profiles were similar for the different sampling locations, where the %OC was highest in the topsoil, decreasing exponentially with depth over the first ~60 cm. At 190 cm depth %OC decreased to 0.1-0.3% for both profiles. For forest transect F1, %OC content in the upper ca. 60 cm was higher at the UM and LM position compared to the other hillslope positions. These differences were particularly apparent in the upper ca ~20 cm of the profile. This difference between the hillslope positions was less marked in F2. However, the %OC was highest for the B-F2, UM-F2and LM-F2 when compared to the V-F2, T-F2 and M-F2."

*Line 210-211: The description of the $\delta^{13}C$ profiles is too brief.*

REPLY: The description of the $\delta^{13}C$ will be improved  by adding few lines as follows:

At the surface, the $\delta^{13}C$ values only showed minor variations, between -27.1 and -25.5‰ (Figure 3a and 3b). $\delta^{13}C$ values increased with depth and reached a value of -24.1 ± 0.6‰ at a depth of 60 cm. Below this depth, $\delta^{13}C$ values no longer showed a systematic variation with depth but varied within a narrow range. No distinct trends were observed for the different sampling positions, where only the $\delta^{13}C$ of F1-T were slightly higher throughout depth when compared to the other sampling positions.

*Line 218-219: It would be better to say that in the first few decimeters, these two profiles have lower SOC values than the other profiles.*

REPLY: Thank you for your suggestion, we will change it accordingly.

*Line 236: The sentence "However, the cumulative…on the GLP hillslope" is unnecessary.*

REPLY: We will remove this.

*Figure 3c: THIS IS NOT THE GOOD ONE!*

REPLY: Thank you for pointing out this error. We will add the correct sub-plot, verify the corresponding text, and check the full manuscript for correct Figure and Table references.

[Figure]

(a) Forest transect F1

(b) Forest transect F2

(c) Grassland transect GLA

(d) Grassland transect GLP

*Discussion*

*Lines 253-277: All these explanations of the evolution of δ 13C values under C3 forest vegetation are excessively long. Since the end of the 80's, many articles have detailed this. This does not provide decisive information to answer the objectives of the paper.*

REPLY: We had elaborated on this topic to provide the reader with the necessary background information to frame the observed decrease in $\delta^{13}C$ that we observed under forest, and to be able to properly compare this with the trends we found under grassland that are described from line 278. However, we agree this might be considered too extensive, and will reduce the length of this part by removing few sentences or summarize some information in the revised version of the manuscript.

*Lines 293-295: I do not agree, in the topsoil (what depth exactly?), the C3 contribution is much lower than 70%! See the figure 6.*

REPLY: This was indeed not clearly formulated, the profile interval we refer to here is the upper 0-50 cm, and we excluded the values in the valley profile. We will clarify this by changing the sentence as follows: " The contribution of C3 plant material to the SOC present in the upper 0-50 cm of these grassland soil profiles is estimated at ca. 70%, with exception of the valley position."

*Lines 297: for GLP-V the δ13C value increases between the surface and 50 cm.*

REPLY: We will correct his.

*Lines 356-358: repetition of the lines 348-350*

REPLY: Thank you for noticing this repetition. Lines 348-350 mainly point out the difference between of erosion which occurred in transects under forest vegetation and grassland vegetation, whereas lines 356-358 refer to differences in erosion between along the transects, i.e. that the erosion rates increase from the top towards the lower slopes. To clarify this, we will combine these 2 sections in the new version of the manuscript as follows: " This is confirmed by soil erosion rates derived from in situ $^{10}Be$ concentrations of the topsoil samples (5-15 cm) which indicates that both under grassland and forest erosion rates increases from the top towards the valley position, where the erosion rates are consistently higher under grassland when compared to forest."

*Line 380: "…, while the outputs include CO2,…" or "…, while the outputs include CO2 emissions,…"?*

REPLY: Thank you for your clarification. We mean here the $CO_2$ emission, it will be corrected as suggested.

*Line 397-398: It is not true that all the studies cited found strong differences in SOC stocks between savannah and forest situations. Moreover, the stocks are not calculated and commented on.*

REPLY: We apologize for the confusion due to missing references - we had intended to refer here to Rabetokotany-Rarivoson et al. (2015) and Razafindrakoto et al., (2018) who have investigated the SOC

change due to land use change by following the different stages of deforestation that occurred in the humid rainforest of Madagascar. They indeed found that the SOC stocks in the soil under the final stage of deforestation (grasses) are always much lower than the SOC stock under the initial forest. We will rephrase this sentences as follow: " Rabetokotany-Rarivoson et al. (2015) and Razafindrakoto et al. (2018) found a strong difference in SOC between the initial forest vegetation and the final stage of deforestation which is characterised by non-forest vegetation (dominated by grasses)".

*Lines 400-401: That is true, but what does it add to the discussion, at this point. It would be better to delete this sentence.*

REPLY: This will be removed.

*Line 411: "The $\delta13C$ values of the forest profiles increased with depth, which is expected for soils developed for soils developed under C3 vegetation". It would be better to say that these 13C profiles are typical of soils under C3 vegetation for a very long time.*

REPLY: Thank you for the suggestion. We will rephrase this sentence as suggested.

*Lines 417-418: you cannot say that organic carbon input from the new grassland vegetation is not significant: it represents almost a third of the carbon stock!*

REPLY: This description might have been somewhat unfortunate - the fraction of SOC from the grass vegetation indeed represents one third of the total SOC stock. What we aimed to communicate here, is that (i) total OC stocks in the grasslands are substantially lower than in forests, and (ii) that despite the absence of substantial new inputs from C3 vegetation, the bulk of the SOC stocks is still largely dominated (70%) by (old) C3-derived carbon.

To clarify our point, we will rephrase this sentence to make this point more clear and avoid misinterpretations.

*Line 429: "This indicates that the response time to deforestation depends on the rate of depletion of the old C3 pool." What does this sentence mean?*

REPLY: What we referred to here is that the time since deforestation is likely to be reflected in the fraction of the C-OC pool that has been mineralized / lost. We agree that the sentence might be unclear for readers and will therefore rephrase this.

*Technical corrections*

*Introduction Line 42: Voarintsoa et al., not Voarintsoa and Cox*

REPLY: Thank you for pointing this out, this will be corrected.

*Materials and methods Figure 1a: in the caption, it is written "dotted black line", but it is a "dotted white line".*

REPLY: This will be corrected in the new version.

*Line 121: The supplementary material S3 does not show vegetation*

REPLY: Thank you for pointing out this error, the correct Figure we should have referred to is S1; this will be corrected.

*Line 148: in the equation, δ13C, not δ13.*

REPLY: This will be corrected.

*Line 197: for D(i), the unit of measurement is missing.*

REPLY: We will add units for D(i) (cm) as well as for the bulk density (g/cm³).

*Results*

*Figure 2: in the caption: "middle" not "middles"*

REPLY: We will correct this.

*Line 207: the topsoil samples are 0-5 cm not 0-10 cm*

REPLY: Will be changed to "in the upper 0-10 cm".

*Line 209: "…between -25.5 and - 27.1‰ …"*

REPLY: We agree that number format should be one number after the decimal point and it should be -27.1 and -25.5‰ (from low to high values). We will change this in the manuscript and keep our number format consistent.

*Line 225-226: verify the profiles which show gradual decline: GLP-B, GLP-UM, GLA-T (not GLA-M)*

REPLY: It will be verified and changed accordingly in the revised manuscript.

*Line 240: "at different depths" appears two times*

REPLY: This will be corrected in the new revised manuscript.

*Line 280: "…values of -20 down…" The symbol ‰ is missing.*

REPLY: The symbol ‰ will be added.

*Line 350: In the references, Brosens et al. is indicated as published in 2022.*

REPLY: The discussed in-situ [10]Be data have not yet been published and are not part of the Brosens et al. (2022) paper. Therefore, we will keep this reference as non-published.

References:

Andriamananjara, A., Ranaivoson, N., Razafimbelo, T., Hewson, J., Ramifehiarivo, N., Rasolohery, A., Andrisoa, R. H., Razafindrakoto, M. A., Razafimanantsoa, M. P., Rabetokotany, N. and Razakamanarivo, R. H.: Towards a better understanding of soil organic carbon variation in Madagascar, Eur. J. Soil Sci., 68(6), 930–940, doi:10.1111/ejss.12473, 2017.

Carroll, C., Merton, L. and Burger, P.: Impact of vegetative cover and slope on runoff, erosion, and water quality for field plots on a range of soil and spoil materials on central Queensland coal mines, Soil Res., 38(2), 313, doi:10.1071/SR99052, 2000.

Cerdan, O., Govers, G., Le Bissonnais, Y., Van Oost, K., Poesen, J., Saby, N., Gobin, A., Vacca, A., Quinton, J., Auerswald, K., Klik, A., Kwaad, F. J. P. M., Raclot, D., Ionita, I., Rejman, J., Rousseva, S., Muxart, T., Roxo, M. J. and Dostal, T.: Rates and spatial variations of soil erosion in Europe: A study based on erosion plot data, Geomorphology, 122(1–2), 167–177, doi:10.1016/j.geomorph.2010.06.011, 2010.

Cerling, T. E. and Harris, J. M.: Carbon isotope fractionation between diet and bioapatite in ungulate mammals and implications for ecological and paleoecological studies, Oecologia, 120(3), 347–363, doi:10.1007/s004420050868, 1999.

Govers, G., Vandaele, K., Desmet, P., Poesen, J. and Bunte, K.: The role of tillage in soil redistribution on hillslopes, Eur. J. Soil Sci., 45(4), 469–478, doi:10.1111/j.1365-2389.1994.tb00532.x, 1994.

Heimsath, A. M., Furbish, D. J. and Dietrich, W. E.: The illusion of diffusion: Field evidence for depth-dependent sediment transport, Geology, 33(12), 949, doi:10.1130/G21868.1, 2005.

Pelletier, J. D. and Rasmussen, C.: Quantifying the climatic and tectonic controls on hillslope steepness and erosion rate, Lithosphere, 1(2), 73–80, doi:10.1130/L3.1, 2009.

Rabetokotany-Rarivoson, N., Andriamananjara, A., Razafimbelo, T., Ramifehiarivo, N., Ramboatiana, N., Razafimanantsoa, M., Razafimahatratra, H., Rabeharisoa, L., Bernoux, M., Brossard, M., Albrecht, A., Winowiecki, L., Vagen, T., Grinand, C., Vaudry, R., Rakotoarijaona, J.-R., Rahagalala, P., Rasolohery, A., Parany, L., Bürren, C., Saneho, H. J., Miasa, E. and Razakamanarivo, H.: Changes in soil organic carbon (SOC) stocks after forest conversion in humid ecoregion of Madagascar, XIV WORLD For. Congr. Durban, South Africa, 7-11 Sept. 2015, (September), 8p, 2015.

Razafindrakoto, M., Andriamananjara, A., Razafimbelo, T., Hewson, J., Andrisoa, R. H., Jones, J. P. G., van Meerveld, I., Cameron, A., Ranaivoson, N., Ramifehiarivo, N., Ramboatiana, N., Razafinarivo, R. N. G., Ramananantoandro, T., Rasolohery, A., Razafimanantsoa, M. P., Jourdan, C., Saint-André, L., Rajoelison, G. and Razakamanarivo, H.: Organic Carbon Stocks in all Pools Following Land Cover Change in the Rainforest of Madagascar, Soil Manag. Clim. Chang. Eff. Org. Carbon, Nitrogen Dyn. Greenh. Gas Emiss., (September 2018), 25–37, doi:10.1016/B978-0-12-812128-3.00003-3, 2018.

Roering, J. J., Kirchner, J. W. and Dietrich, W. E.: Evidence for nonlinear, diffusive sediment transport on hillslopes and implications for landscape morphology, Water Resour. Res., 35(3), 853–870, doi:10.1029/1998WR900090, 1999.

Silburn, D. M., Carroll, C., Ciesiolka, C. A. A., DeVoil, R. C. and Burger, P.: Hillslope runoff and erosion on duplex soils in grazing lands in semi-arid central Queensland. I. Influences of cover, slope, and soil, Soil Res., 49(2), 105–117, doi:10.1071/SR09068, 2011.

Zhao, M., Jacobs, L., Bouillon, S. and Govers, G.: Rapid soil organic carbon decomposition in river systems: effects of the aquatic microbial community and hydrodynamical disturbance, Biogeosciences, 18(4), 1511–1523, doi:10.5194/bg-18-1511-2021, 2021.

---

## Author Comment (AC2)

**Author response_referee#1**

We would like to thank reviewer #1 for the very constructive and thorough feedback on our manuscript. We appreciate all suggestions, which will improve the quality and rigour of our manuscript. Below, we listed the main changes to the manuscript as a reply to the primary concerns raised by the reviewer

- We will enhance the description of the study area:
  - Description of the lateritic zone where we collected all our soil samples
  - Texture of soil and topographical characteristics.
  - Short overview of C3 grasses location and expansion in Madagascar.
- The effect of erosion resulting from vegetation changes will be discussed in more detail
- Background on the use of $^{13}$C and $^{14}$C on will be added in the introduction.
- We will insert additional supplementary figures and statistical analyses, in particular relating to the difference between $\delta^{13}$C-OC or OC content of the valley and other profile positions.

Reviewer comments are indicated in *italics*, our responses in regular font**.**

*In this paper the authors present SOC concentration and stock, 13C, and 14C depth profiles from hillslope transects with forest and grassland vegetation cover in the highlands of Madagascar. The authors use these data to address a debated question – whether current grasslands are grasslands because of bioclimatic and edaphic factors (ie. they are "natural" grasslands") or if they are the consequence of deforestation by humans hundreds of years ago. They argue that 13C depth profiles indicate a shift from C3 (possibly forest) to C4 (current grasses). They further argue that conversion from forest to grassland has caused the sustained loss of SOC since this time as current grasslands store about half as much carbon as intact forests. These data and findings are interesting, but I find that the manuscript could use some improvements and corrections prior to publication.*

REPLY: First, we would like to thank referee #1 for his/her positive evaluation and comments on our manuscript. All comments are helpful to improve the presentation of our results and manuscript.

Reviewer #1 General concerns are as follows:

1) *Though there is some consideration of erosion, this could be better explained and addressed in the abstract and discussion sections. This needs to be fully considered as an alternative explanation to the differences in SOC especially considering the presence of gully erosion (lavaka) and lateritic horizons in some grassland areas.*

REPLY: Indeed, erosion rates are considered to be higher after vegetation change (from forest to grasslands), and this, therefore, could contribute to the higher $\delta^{13}$C and OC at the valley position compared to the upper hillslope position for the grassland profiles (Top - Upper middle-Middle – Lower middle and Bottom). The effect of erosion will be discussed in more detail in the Discussion of the revised manuscript.

*2) There is no other discussion of alternative sources of carbon. At least indicate you've considered carbonate and geogenic OC. What would their presence mean for your findings and conclusions? Why do you think you do not need to consider them?*

REPLY: It should first be noted that the data presented only refer to organic carbon: all carbonates were eliminated when preparing our soil for analysis by acidification after weighing subsamples in Ag cups (see Materials & Methods). According to the World Reference Base for Soil Resources, the soils in our study area are classified as ferralsols (WRB, 2006). In addition, the basement rocks on the site we sample our soil are metamorphic and igneous (Du Puy and Moat, 1996). Therefore, we did not consider geogenic OC to be substantial, in contrast to subsoils developed from sedimentary rocks where this might be more important (Graz et al., 2010). We will mention this explicitly in the revised version.

*3) Inferring that the conversion to grassland is what caused the large discrepancy in SOC stocks between the grassland and forest transects is interesting but requires consideration of how similar or different the soils are independent of the vegetation cover now – if erosion is a factor now, could it have been before when the vegetation was C3 dominated according to the $^{13}$C results? Why do you think they were similar? Are the textures similar?*

REPLY: We agree that our soil should be comparable before testing our hypothesis: soil type, topography, slope gradient, and texture. We will mention in the manuscript that we did not observe significant

differences in the texture of soils under grassland and forest. We will add a new figure of texture and gradient as a supplementary Figure. Regarding the possible effect of erosion under C3 vegetation: the $\delta^{13}$C data collected along our forest transects do not show substantial differences according to the position along the hillslope. This suggests that erosion might not play a major role in the variation of $\delta^{13}$C in a C3 (in this case, forest) dominated landscape as significant erosion would invariably have led to sign of OC accumulation in the valley position. Water erosion is likely to be more important under grassland: this explains the fact that there is clear accumulation of SOC in the valley positions under grassland. This finding is not surprising: water erosion rates under a dense forest cover are generally very low (see, e.g., Cerdan et al. (2010) who reported an average water erosion rate of 0.14 t ha$^{-1}$y$^{-1}$ under forest in Europe and Zhao et al., (2021) who reported a median erosion rate of 0.15 t ha$^{-1}$y$^{-1}$ and an average erosion rates of 1.5 t ha$^{-1}$y$^{-1}$ under forest in China).

*4) is it possible that previous C3 vegetation may not have been forest (possibly savanna or C3 grassland, which is common in other parts of the tropics)?*

REPLY: C3 endemic grass species that has been inventoried in Madagascar belong to the "forest shade clade" (Paniceae: Boivinellinea) and bamboos (Hackel et al., 2018). Their diversification since the Miocene is reported to be favored by the expansion of the *Sambirano* rainforest (in the North of Madagascar) (Hackel et al., 2018; Yoder and Nowak, 2006). Therefore, if C3 grasses had existed in our study area, it would have been within a forest ecosystem. We will clarify this in the revised version of our manuscript.

*5) The introduction needs some background on the use of 13C and 14C in this context (for vegetation shifts and erosion) as well as context for why these differences in SOC stocks are relevant. There is a lot of good literature on the impacts of agriculture (from the beginning of agriculture, not limited to contemporary studies) on SOC to draw from here.*

REPLY: Thank you for this suggestion; a background paragraph on the use of $^{13}$C and $^{14}$C will be added in the introduction paragraph and we will make sure that the impact of deforestation/conversion to grassland as presented in the literature is included.

*6) there are no statistical analyses included in this manuscript. It seems the work could benefit from some relatively simple correlation, regression, and ANOVA to address whether it is appropriate to average all of the hillslope positions, for example. Is there no difference in the valleys or are the valleys just more similar than the other hill slope positions?*

REPLY: We agree that statistical analysis will strengthen our conclusions. A summary table showing the result of the statistical analyses will be added in the supplementary files. We will compare data from the valley sites to the other hillslopes positions (in the current grassland) using simple significant differences tests and will thereby concentrate on aggregate measures, such as the total SOC stock down to a specific depth or the average $^{13}$C signature. More specifically we will answer/illustrate the following questions using statistical analysis:

- Are the profiles under forest and under grassland significantly different with respect to SOC content and signature ?
- Are the valley positions significantly different from the hillslope positions (i) under forest and (ii) under grassland ?.

Specific comments from referee #1:

*-L26: what about geogenic C, which could have a 13C value similar to C3 vegetation. How confident are you this is trees and not C3 savanna or grassland?*

REPLY:  As outlined in response to previous general comments: in Madagascar, C3 grasses had existed only within a forest ecosystem; all open grasslands are characterized by C4 vegetation. Geogenic OC is not considered as a source of OC because the basement rocks are metamorphic and igneous, and all carbonates were eliminated when preparing our soil for analysis.

*-L31: What do you mean by "recent expansion" and why do you think this is 1) recent and 2) expansion? Why not just high productivity in the valleys or erosional deposition of C from the surface up slope? This would also explain why the SOC stocks in the valleys are so high and similar to the forest more so than a recent expansion (I think, but maybe I am missing something?)*

REPLY: We do agree with the reviewer that these mechanisms may also be important in explaining the characteristics of the valley profiles and we will change our wording to include these mechanisms as possible explanation for the high carbon content of the valley and the young age of the SOC at the valley floor positions.  The sentence then becomes as follows:

"At the valley positions under grassland the upper 80 cm topsoil contains larger amounts of recent, grass-derived OC in comparison to the hillslope positions. This is likely to be  related to the higher productivity of the valley grasslands (due to higher moisture and nutrient availability) but deposition of OC that was eroded further upslope may also have contributed".

*-L75: a word is missing here "do not allow assess how"*

REPLY: We will change to "allow us to assess".

*-L85-6: 13C, 14C, and SOC stock relevance need to be presented earlier in the introduction.*

REPLY: A background on the use and relevance of $^{13}C$ and $^{14}C$ data, as well as SOC stocks will be added to the introduction.

*-L93: again, a word is missing here "allow to assess"*

REPLY:  We will change this as suggested.

*-L105: If at many locations there are lateritic horizons, you need to indicate whether you sampled in any of these areas later. What does this mean for your findings?*

REPLY: The lateritic soil horizon is usually between 0.5 to 2m thick (Voarintsoa et al., 2012). Our soil samples have been sampled in the lateritic soil horizon. We will indicate this in the material and methods sections in the revised version. The lateritic soil horizon is considered to be relatively impermeable, thus favoring surface runoff erosion, especially if there is no or little vegetation  and if no cracks are present (Wells and Andriamihaja, 1993). However, it should also be pointed out that, while there was a lateritic soil horizon, a true laterite was not present at our sites.

*-L108: lavaka need to be better explicitly addressed in the context of erosion in the current grassland areas – what impact can their presence have on your results? How old are they – do they predate human deforestation or are they possibly a consequence of humans using these areas for grazing? Land cover conversion and land use may be conflated here or not independently addressed adequately. They seem used interchangeably.*

REPLY: An important point is that we did not sample inside lavaka, we consider the presence of lavaka but on hillslopes outside of the lavaka. By choosing a slope with and without lavaka we wanted to investigate whether soils on slopes that have lavaka development may differ from slopes that do not have them. We found that the OC content and $^{13}C$ value do not differ significantly (at the surface) and have the same trend with depth for GLP and GLA. A statistical analysis of this will be provided in the supplementary file. We will explain this better in the revised version of the manuscript.

Previous research has shown that some lavaka can be directly associated with human activities such as trenches, tracks, steep fields and the construction of canals and paddies (Riquier, 1954; Wells and Andriamihaja, 1993). However, other lavaka are tens of thousands of years old, predating the permanent settlement of humans in the highlands, which is estimated to have taken place between 1600 and 100 years ago (Douglass et al., 2019; Mietton et al., 2014; Wells and Andriamihaja, 1993). Recent research has shown that lavaka in the Lake Alaotra region are on average ca. 400 years old. Lavaka became far more numerous since ca. 1000 years ago and lavaka formation rates have increased dramatically over the last 200 years. This timing and the rapid increase in lavaka erosion rates has been confirmed by floodplain sedimentation data in the same area (Brosens et al., 2022). Brosens et al. (2022) links this increase in lavaka erosion to increased environmental pressure due to growing human populations and intensified grazing based on scenario modelling and on the absence of significant climatic variations in the period considered. The mechanisms that lead to the initiation of lavaka, which typically occurs at the hillslope between upper middle and lower middle position, are not well understood. Different theories have been developed to explain their initiation, where both surface runoff processes and groundwater sapping are hypothesized to play an important role (Wells and Andriamihaja, 1993). However, the fact that excessive pressure on the land plays a critical role in lavaka initiation suggests that changes in surface properties related to overgrazing/overuse such as soil compaction and the decrease in vegetation cover and the increase of surface runoff caused by these changes play a crucial role. We will briefly re-iterate the main findings of Brosens et al. (2021) in the revised version of the manuscript.

*Table 1: Reported errors are > 1 so you should not report decimal places as they are within your uncertainty. Is it appropriate to present the data this way by averaging across landscape position? The presence of a large difference between the forests and grasslands except in the valleys suggests that maybe it is not appropriate as does the statement that the grassland hillslopes may be different from one another. This table is redundant with Figure 4, isn't it? The figure is much more informative, and you provide these values in the text – they do not need to be reported in the main paper so many times. If you find value in the table, move it to the supplement.*

REPLY: The table will be presented in the supplementary figure, and the number of decimals will be adjusted. We do indeed average across landscape positions. Given that profiles at different landscape positions are very similar and that their variation is not in any way related to landscape position, we believe this is justified. Also the data are simply used to make a comparison of SOC inventories under forest vs grassland and we believe that by presenting the data in this way this comparison can be most easily made.

*L345-6: this suggests the surface young C has been eroded, which would explain why the valley has more SOC and younger C but this does not seem adequately discussed as an important part of the story for the grassland transects.*

REPLY: As suggested, a section of the effect erosion that follows the vegetation change will be added to grassland transect discussions and we will refer to the effects of erosion as a possible explanation.

*L269: This paragraph is correct but the way the logic is presented is a little confused in my opinion. Important to this explanation but only implied, is that respiration would be depleted in $^{13}C$ relative to the organic matter because the light isotope is preferentially converted to CO2 and diffused to the surface – this is based on mass-dependent fractionation and is why the heavy isotope remains behind in the microbial biomass and byproducts. This is why the leaves that are taking up CO2 in soil respiration may be depleted relative to leaves taking up CO2 from well mixed air higher in the canopy. Also important is that mass-dependent fractionation causes the light isotope to be transported within the plants, so roots and root respiration are also quite depleted in 13C relative to the classic values for C3 plant leaf tissue of -25 permil. Similarly within a tree leaves growing closer to the ground may be more depleted that leaves in the upper canopy.*

REPLY: As the reviewer explains, the understory effect or canopy effect found in tropical forest is mainly related to the gradient in $\delta^{13}C$ of ambient $CO_2$ along the vertical gradient: $\delta^{13}C$-$CO_2$ is low close to the ground, due to elevated $CO_2$ concentrations via the contribution of soil respiration. Higher up in the canopy, the $\delta^{13}C$-$CO_2$ values are closer to the average atmospheric $CO_2$ composition. We will try to express this more clearly in the revised version.

*L278: Figures 3a and 3c look the same. Only 3 d looks like it may be different. Is this a mistake? There are no statistics again to assess what differences are statistically significant, making ecological significance questionable.*

REPLY: We apologize for this error; the right figure will be corrected (see below). Statistical analyses will be provided in the supplementary file in the revised version.

[Figure]

(a) Forest transect F1

(b) Forest transect F2

(c) Grassland transect GLA

(d) Grassland transect GLP

*L296: Could this be because of deposition from soil that originated upslope via erosion? Could this explain why topsoils don't have more enriched 13C values on the slopes?*

REPLY: We consider all alternatives that might explain the higher value of $\delta^{13}C$ in the valley. We suggested that the value of $\delta^{13}C$ of the top position could be due to vegetation change, which induces more erosion in the upper slope positions. In addition, there is a higher vegetation density in the valley that we observe compared to other profiles positions. We will clearly express this in the revised version of the MS.

*L355: Rephrase for clarity – something like "Surface erosion is expected to be variable across topographic positions along the hillslope transect, with minimal…."*

REPLY: Thank you for this suggestion; the sentences will be changed as suggested.

*L357: what does "10Be in-situ topsoil samples" mean? I am more familiar with "in situ 10Be" which means cosmogenic formation of 10Be when surfaces are exposed in rock or sediment. This is an analysis so again this phrasing does not make sense to me. Try "erosion rates from in-situ 10Be analysis of the topsoil samples" perhaps? Also, please clarify what you would expect in terms of variation in the pMC and $^{13}$C values based on the erosion rates indicated by the $^{10}$Be analyses.*

REPLY: We apologize for the confusion; the sentences will be modified as suggested. With the data that we have now, we could not really have a specific expectation for pMC and $^{13}$C values based on the erosion rates indicated by the 10Be analyses. However, the fact that we saw an increase in OC, $\delta^{13}$C and pMC values at the valley-position under grasslands seems to indicate that at this position soil that has been eroded from upslope position is deposited. This is not observed in the forest transect, which is consistent with low erosion rates, with minimal deposition taking place at the valley position.

*L358-9: This is very hard to see in figure 5. It is much easier to see in figure 3 for the $^{13}$C. Please provide a similar figure as Figure 3 to show the 14C value. If it is only useful for this statement, put it in the supplement. I would very much appreciate seeing this figure along with the depth profiles for SOC and 13C as you have shown.*

REPLY: Thank you for this suggestion, we will include a new figure (see below) in the supplementary figures:

[Figure]

New supplementary Figure: Depth profile of pMC for GLP-T, GLP-V, F2-T and F2-V.

*L360: Again, some statistics would be great and could strengthen your story. Correlation or regression would be very simple but quantify the relationship you see between the isotopes in the grassland transects. On figure 5, drawing a regression line on this plot for the grasslands (and also perhaps for the forests) would also drive home your point about how the grassland values converge with the forest ones at depth and make it easier to identify the depth labels on the different datapoints, which are quite difficult to read. Also, figure 5 would be easier to digest if the grassland points had the same symbols and color, with one open and one closed like the forests. This would make the figure feel less cluttered and make it easier to*

*pick out the labels for the depths and transects. I am unsure why the hillslopes and valleys are marked using different symbols – I do not see a pattern. Is there one? If so this plot should be further improved to make it easier to see. I see the grasslands falling on one regression line and the forests on another.*

REPLY: Thank you for this suggestion. We used different symbols for the valley to show the difference between pMC value found in the valley position of forest and grassland transect. In this figure 5, we highlighted that the valley of grassland at the upper 50 cm is composed of modern OC (pMC≥100%), which is not the case of the OC in the valley position of the forest. In addition, we also highlighted that there is a pMC trend; the grassland is reaching the pMC value of the forest SOC at the subsoil. Therefore, this figure will be improved as suggested, and a regression line will be added.

*L365-378 This section should be significantly shortened to just a few sentences about how your findings are similar to other similar studies. There is no introduction or context about why the stocks or distribution of stocks are important so it is very out of place in a manuscript so focused on vegetation shifts and erosion across hillslopes. What about how similar these soils were prior to when humans may have deforested the current grasslands? What else could explain your results? What about the laterite? What about the lavaka – when did it form and what influence does it have on your findings? What other things may explain your findings other than human deforestation? I very much like the suggestion in this section that there may be long sustained loss of C and I think this is consistent with what long term global evidence for a massive loss of C since the dawn of agriculture has been. But this needs to be better substantiated in the paper through consideration of alternative explanations.*

REPLY: Thank you for this suggestion, as outlined in response to other comments, we will (i) introduce the importance of SOC stocks in the introduction, and (ii) add more context on the laterite depth and lavaka.

*Figure 6: Is averaging the grassland profiles like this valid? There is no effect of the lavaka? Are some of these sites influenced by laterite?*

REPLY: We collected all our soil samples on the hillslope and did not find any significant difference between the hillslope with and without lavaka: we will demonstrate this statistically in the revised version of the Material and Methods sections. The two grassland transects that we analyzed do not show any statistical difference in terms of δ$^{13}$C and OC content at the surface, and they show the same trend with depth. We therefore think it is justified to combine the data here.

*Figure 7: This is redundant with figure 3, no? chose which one best shows your results (I think figure 3 but it is difficult to tell as it seems to have a mistake). If you like both plot types, move the less impactful one to the supplement.*

REPLY: Figure 3 mainly compares the value in each transect. Figure 7 compares δ$^{13}$C of forest and grassland for each position and shows that δ$^{13}$C become similar at a lower depth. We agree that the underlying results we present are the same – but feel it is still useful to present them both ways; we will therefore move Figure 7 to the supplement as suggested.

References

Brosens, L., Broothaerts, N., Campforts, B., Jacobs, L., Razanamahandry, V. F., Van Moerbeke, Q., Bouillon, S., Razafimbelo, T., Rafolisy, T. and Govers, G.: Under pressure: Rapid lavaka erosion and

floodplain sedimentation in central Madagascar, Sci. Total Environ., 806, 150483, doi:10.1016/j.scitotenv.2021.150483, 2022.

Cerdan, O., Govers, G., Le Bissonnais, Y., Van Oost, K., Poesen, J., Saby, N., Gobin, A., Vacca, A., Quinton, J., Auerswald, K., Klik, A., Kwaad, F. J. P. M., Raclot, D., Ionita, I., Rejman, J., Rousseva, S., Muxart, T., Roxo, M. J. and Dostal, T.: Rates and spatial variations of soil erosion in Europe: A study based on erosion plot data, Geomorphology, 122(1–2), 167–177, doi:10.1016/j.geomorph.2010.06.011, 2010.

Douglass, K., Hixon, S., Wright, H. T., Godfrey, L. R., Crowley, B. E., Manjakahery, B., Rasolondrainy, T., Crossland, Z. and Radimilahy, C.: A critical review of radiocarbon dates clarifies the human settlement of Madagascar, Quat. Sci. Rev., 221, doi:10.1016/j.quascirev.2019.105878, 2019.

Graz, Y., Di-Giovanni, C., Copard, Y., Laggoun-Défarge, F., Boussafir, M., Lallier-Vergès, E., Baillif, P., Perdereau, L. and Simonneau, A.: Quantitative palynofacies analysis as a new tool to study transfers of fossil organic matter in recent terrestrial environments, Int. J. Coal Geol., 84(1), 49–62, doi:10.1016/j.coal.2010.08.006, 2010.

Hackel, J., Vorontsova, M. S., Nanjarisoa, O. P., Hall, R. C., Razanatsoa, J., Malakasi, P. and Besnard, G.: Grass diversification in Madagascar: In situ radiation of two large C 3 shade clades and support for a Miocene to Pliocene origin of C 4 grassy biomes, J. Biogeogr., 45(4), 750–761, doi:10.1111/jbi.13147, 2018.

Mietton, M., Cordier, S., Frechen, M., Dubar, M., Beiner, M. and Andrianaivoarivony, R.: New insights into the age and formation of the Ankarokaroka lavaka and its associated sandy cover (NW Madagascar, Ankarafantsika natural reserve), Earth Surf. Process. Landforms, n/a-n/a, doi:10.1002/esp.3536, 2014.

Du Puy, D. J. and Moat, J.: A refined classification of the primary vegetation of Madagascar based on the underlying geology: using GIS to map its distribution and to assess its conservation status, Biogéographie de Madagascar, 205–218, 1996.

Riquier, J.: Etude sur les Lavaka, Mem. l'Institut Sci. Madagascar, D(VI), 169–189, 1954.

Voarintsoa, N. R. G., Cox, R., Razanatseheno, M. O. M. and Rakotondrazafy, A. F. M.: Relation between bedrock geology, topography and lavaka distribution in Madagascar, South African J. Geol., 115(2), 225–250, doi:10.2113/gssajg.115.225, 2012.

Wells, N. A. and Andriamihaja, B.: The initiation and growth of gullies in Madagascar: are humans to blame?, Geomorphology, 8(1), 1–46, doi:10.1016/0169-555X(93)90002-J, 1993.

WRB: World Reference Base for Soil Resources (WRB), in Encyclopedia of Environmental Change, SAGE Publications, Ltd., 2455 Teller Road, Thousand Oaks, California 91320., 2006.

Yoder, A. D. and Nowak, M. D.: Has Vicariance or Dispersal Been the Predominant Biogeographic Force in Madagascar ? Only Time Will Tell, , doi:10.1146/annurev.ecolsys.37.091305.110239, 2006.

Zhao, M., Jacobs, L., Bouillon, S. and Govers, G.: Rapid soil organic carbon decomposition in river systems: effects of the aquatic microbial community and hydrodynamical disturbance, Biogeosciences, 18(4), 1511–1523, doi:10.5194/bg-18-1511-2021, 2021.

---

## Author Response (AR1)

**Responses to reviewers' comments**

We thank both reviewer for their constructive comments and suggestions which allowed us to improve our methodological description, the overall manuscript structure, and the data analysis/interpretation. Below we have listed the main changes we made in the revised version of our manuscript:

- We extended the introduction section, by adding some background on the use $^{14}$C and $^{13}$C to study the impact of deforestation or vegetation changes.
- We provided more detailed information on our study area and our soil characteristics:
    - Lavaka and erosion processes
    - Lateritic horizon
    - Type of soil and soil rock basement
    - Topographical characteristics (slope length, elevation and gradient) of the forested and grassland transects
    - soil texture data are now included
    - The occurrence of endemic C3 grasses
- The discussion section was improved based on the comments received, and new statistical analyses are included.
- Inconsistencies in figure captions, references, equations and units were corrected.

These changes are described in detail below, where you can find our point-by point response to the reviewer 'comments. For clarity, the comments of the reviewers are in *italics* while our response is given normal font, with an indication of the lines and references in the revised MS track-change indicated in **bold.**

**Author response referee#1**

*In this paper the authors present SOC concentration and stock, 13C, and 14C depth profiles from hillslope transects with forest and grassland vegetation cover in the highlands of Madagascar. The authors use these data to address a debated question – whether current grasslands are grasslands because of bioclimatic and edaphic factors (ie. they are "natural" grasslands") or if they are the consequence of deforestation by humans hundreds of years ago. They argue that 13C depth profiles indicate a shift from C3 (possibly forest) to C4 (current grasses). They further argue that conversion from forest to grassland has caused the sustained loss of SOC since this time as current grasslands store about half as much carbon as intact forests. These data and findings are interesting, but I find that the manuscript could use some improvements and corrections prior to publication.*

REPLY: First, we would like to thank referee #1 for his/her positive evaluation and comments on our manuscript. All comments were helpful to improve the presentation of our results and manuscript.

*Reviewer #1 General concerns are as follows:*

1) *Though there is some consideration of erosion, this could be better explained and addressed in the abstract and discussion sections. This needs to be fully considered as an alternative explanation to the differences in SOC especially considering the presence of gully erosion (lavaka) and lateritic horizons in some grassland areas.*

REPLY: Indeed, erosion rates are considered to be higher after a change in vegetation (from forest to grasslands), and this, therefore, could contribute to the higher $\delta^{13}C$ and OC at the valley position compared to the upper hillslope position for the grassland profiles (Top - Upper middle-Middle – Lower middle and Bottom). We have now mentioned this in the abstract and discussed the possible effect of erosion in more detail in the revised Discussion.

**Abstract. L31-38.**
**4.3 Effect of erosion following natural vegetation change in the grassland transects**. **L620-633.**

*2) There is no other discussion of alternative sources of carbon. At least indicate you've considered carbonate and geogenic OC. What would their presence mean for your findings and conclusions? Why do you think you do not need to consider them?*

REPLY: It should first be noted that the data presented only refer to organic carbon: all carbonates were eliminated when preparing our soil for analysis, by acidification after weighing subsamples in Ag cups (see Materials & Methods, L203-204). According to the World Reference Base for Soil Resources (2006), the soils in our study area are classified as ferralsols. In addition, the basement rocks on the site we sample our soil are metamorphic and igneous (Du Puy and Moat, 1996). Therefore, we did not consider geogenic OC to be substantial, in contrast to subsoils developed from sedimentary rocks where this might be more important (Graz et al., 2010). We now mentioned this explicitly in the revised version.

**2.1 Study area: Lake Alaotra region. L140; L152-154.**

*3) Inferring that the conversion to grassland is what caused the large discrepancy in SOC stocks between the grassland and forest transects is interesting but requires consideration of how similar or different the soils are independent of the vegetation cover now – if erosion is a factor now, could it have been before*

*when the vegetation was C3 dominated according to the ¹³C results? Why do you think they were similar? Are the textures similar?*

REPLY: We agree that our soils should be comparable before testing our hypothesis: soil type, topography, slope gradient, and texture. We have now mentioned in the manuscript that we did not observe significant differences in the texture of soils under grassland and forest. We added a new figure presenting the soil texture data and the topographical gradient as a supplementary Figure (S3 and S5). Regarding the possible effect of erosion under C3 vegetation: the $\delta^{13}$C data collected along our forest transects do not show substantial differences according to the position along the hillslope. This suggests that erosion might not play a major role in the variation of $\delta^{13}$C in a C3-dominated landscape (in this case, forest) as significant erosion would invariably have led to signs of OC accumulation in the valley position. Water erosion is likely to be more important under grassland: this explains the fact that there is clear accumulation of SOC in the valley positions under grassland. This line of discussion has been added to the revised manuscript.

**2.2 Sampling transect. L178-180,**
**2.3.3 Soil texture analyses : L269-L275,**
**3.1 Soil texture for grassland and forest soils. L286-L296,**
**4.3 Effect of erosion following natural vegetation change in the grassland transects. L620-630.**

*4) is it possible that previous C3 vegetation may not have been forest (possibly savanna or C3 grassland, which is common in other parts of the tropics)?*

REPLY: Endemic C3 grass species that have been inventoried in Madagascar belong to the "forest shade clade" (Paniceae: Boivinellinea) and bamboos (Hackel et al., 2018). Their diversification since the Miocene is reported to be favored by the expansion of the *Sambirano* rainforest (in the North of Madagascar) (Hackel et al., 2018; Yoder and Nowak, 2006). Therefore, if C3 grasses had existed in our study area, it would have been within a forest ecosystem. We clarified this in the revised version of our manuscript.

**2.1 Study area: Lake Alaotra region. L162-L165**

*5) The introduction needs some background on the use of 13C and 14C in this context (for vegetation shifts and erosion) as well as context for why these differences in SOC stocks are relevant. There is a lot of good literature on the impacts of agriculture (from the beginning of agriculture, not limited to contemporary studies) on SOC to draw from here.*

REPLY: Thank you for this suggestion;  a background paragraph on the use of ¹³C and ¹⁴C has been added in the introduction and the impact of deforestation/conversion to grassland as presented in the literature is now also included.

**Introduction. L96-L106.**

*6) there are no statistical analyses included in this manuscript. It seems the work could benefit from some relatively simple correlation, regression, and ANOVA to address whether it is appropriate to average all of the hillslope positions, for example. Is there no difference in the valleys or are the valleys just more similar than the other hill slope positions?*

REPLY: A summary table showing the results of the statistical analyses has been added in the supplementary materials, and we refer to this accordingly in the revised manuscript.

**2.4 Statistical analyses. L276-284**

Specific comments from referee #1:

*-L26: what about geogenic C, which could have a 13C value similar to C3 vegetation. How confident are you this is trees and not C3 savanna or grassland?*

REPLY: As outlined in response to previous general comments: in Madagascar, C3 grasses had existed only within a forest ecosystem; all open grasslands are characterized by C4 vegetation. Geogenic OC is not considered as a significant source of OC because the basement rocks are metamorphic and igneous, and all carbonates were eliminated when preparing our soil for analysis.

**2.1 Study area: Lake Alaotra region. L152-154; L162-164.**

*-L31: What do you mean by "recent expansion" and why do you think this is 1) recent and 2) expansion? Why not just high productivity in the valleys or erosional deposition of C from the surface up slope? This would also explain why the SOC stocks in the valleys are so high and similar to the forest more so than a recent expansion (I think, but maybe I am missing something?)*

REPLY: We agree with the reviewer that these mechanisms may also be important in explaining the characteristics of the valley profiles and our wording has been changed to include these mechanisms as a possible explanation for the high OC content of the valley soils and the young SOC age there.. The sentence has been changed as follows:

"At the valley positions under grassland, the upper 80 cm contains higher amounts of recent grass-derived OC in comparison to the hillslope positions. This is likely to be related to the higher productivity of the grassland valleys (due to higher moisture and nutrient availability), and deposition of OC that was eroded further upslope may also have contributed".

**Abstract. L30-L38**

*-L75: a word is missing here "do not allow assess how"*

REPLY: We changed to "do not allow us to assess".

**Introduction. L93**

*-L85-6: 13C, 14C, and SOC stock relevance need to be presented earlier in the introduction.*

REPLY: Some background on the use and relevance of $^{13}C$ and $^{14}C$ data, as well as SOC stocks have been added to the introduction.

**Introduction. L96-L106**

*-L93: again, a word is missing here "allow to assess"*

REPLY:  We have corrected this as suggested.

**Introduction. L121**

 *-L105: If at many locations there are lateritic horizons, you need to indicate whether you sampled in any of these areas later. What does this mean for your findings?*

REPLY: The lateritic soil horizon is usually between 0.5 to 2m thick (Voarintsoa et al., 2012). Our soil samples have been sampled in the lateritic soil horizon, this is now specified in the revised Material and Methods sections. The lateritic soil horizon is considered to be relatively impermeable, thus favoring surface runoff erosion, especially if there is no or little vegetation  and if no cracks are present (Wells and Andriamihaja, 1993). These have been clarified in the revised manuscript.

**2.1 Study area: Lake Alaotra region. L143-154**
**2.2 Sampling transect. L173**

*-L108: lavaka need to be better explicitly addressed in the context of erosion in the current grassland areas – what impact can their presence have on your results? How old are they – do they predate human deforestation or are they possibly a consequence of humans using these areas for grazing? Land cover conversion and land use may be conflated here or not independently addressed adequately. They seem used interchangeably.*

REPLY: An important point is that we did not sample inside lavaka, we consider the presence of lavaka but sampled along a transect outside of the actual lavaka present. By choosing a slope with and without lavaka we wanted to investigate whether soils on slopes that have lavaka development may differ from slopes that do not have them. We found that the OC content and $\delta^{13}C$ values did not differ significantly (at the surface) and our profiles showed similar trend for GLP and GLA. A statistical analysis of this is now provided. In addition to the lavaka and their development, we have clarified their initiation process in the revised version of the manuscript.

Previous research has shown that some lavaka can be directly associated with human activities such as trenches, tracks, steep fields and the construction of canals and paddies (Riquier, 1954; Wells and Andriamihaja, 1993). However, other lavaka are tens of thousands of years old, predating the permanent settlement of humans in the highlands, which is estimated to have taken place between 1600 and 100 years ago (Douglass et al., 2019; Mietton et al., 2014; Wells and Andriamihaja, 1993). Recent research has shown that lavaka in the Lake Alaotra region are on  average ca. 400 years old. Lavaka became far more numerous since ca. 1000 years ago and lavaka formation rates have increased dramatically over the last 200 years. This timing and the rapid increase in lavaka erosion rates has been confirmed by floodplain sedimentation data in the same area (Brosens et al., 2022). Brosens et al. (2022) links this increase in lavaka erosion to increased environmental pressure due to growing human populations and intensified grazing based on scenario modelling and on the absence of significant climatic variations in the period considered. The mechanisms that lead to the initiation of  lavaka, which typically occurs at the hillslope between upper middle and lower middle position, are not well understood. Different theories have been developed to explain their initiation, where both surface runoff processes and groundwater sapping are hypothesized to play an important role (Wells and Andriamihaja, 1993). However, the fact that excessive

pressure on the land plays a critical role in lavaka initiation suggests that changes in surface properties related to overgrazing/overuse such as soil compaction and the decrease in vegetation cover and the increase of surface runoff caused by these changes play a crucial role. We have briefly re-inserted the main findings of Brosens et al. (2021) in the revised version of the manuscript.

**2.1 Study area: Lake Alaotra region. L134-139**

*Table 1: Reported errors are > 1 so you should not report decimal places as they are within your uncertainty. Is it appropriate to present the data this way by averaging across landscape position? The presence of a large difference between the forests and grasslands except in the valleys suggests that maybe it is not appropriate as does the statement that the grassland hillslopes may be different from one another. This table is redundant with Figure 4, isn't it? The figure is much more informative, and you provide these values in the text – they do not need to be reported in the main paper so many times. If you find value in the table, move it to the supplement.*

REPLY: The table is now presented in the supplementary material, and the number of decimals has been adjusted. We do indeed average across landscape positions. Given that profiles at different landscape positions are very similar and that their variation is not in any way related to landscape position, we believe this is justified. Also the data are simply used to make a comparison of SOC inventories under forest vs grassland and we believe that this approach is suitable for such a comparison.

**3.4 Soil organic carbon stocks for grassland and forest soil profiles. L361-364**
**Supplementary table 3**

*L345-6: this suggests the surface young C has been eroded, which would explain why the valley has more SOC and younger C but this does not seem adequately discussed as an important part of the story for the grassland transects.*

REPLY: As suggested, a section on the effect erosion that follows the vegetation change has been added to the discussion on the grassland transect data.

**4.3 Effect of erosion following natural vegetation change in the grassland transects. L620-630.**

*L269: This paragraph is correct but the way the logic is presented is a little confused in my opinion. Important to this explanation but only implied, is that respiration would be depleted in $^{13}$C relative to the organic matter because the light isotope is preferentially converted to CO2 and diffused to the surface – this is based on mass-dependent fractionation and is why the heavy isotope remains behind in the microbial biomass and byproducts. This is why the leaves that are taking up CO2 in soil respiration may be depleted relative to leaves taking up CO2 from well mixed air higher in the canopy. Also important is that mass-dependent fractionation causes the light isotope to be transported within the plants, so roots and root respiration are also quite depleted in 13C relative to the classic values for C3 plant leaf tissue of -25 permil. Similarly within a tree leaves growing closer to the ground may be more depleted that leaves in the upper canopy.*

REPLY: As the reviewer explains, the understory effect or canopy effect found in tropical forest is mainly related to the gradient in $\delta^{13}$C of ambient $CO_2$ along the vertical gradient: $\delta^{13}$C-$CO_2$ is lower close to the ground, due to elevated $CO_2$ concentrations via the contribution of soil respiration. Higher up in the

canopy, the $\delta^{13}C$-$CO_2$ values are closer to the average atmospheric $CO_2$ composition. We have tried to express this more clearly in the revised version.

**4.1 Difference in carbon sources between grassland and forest soils. L423-426**

*L278: Figures 3a and 3c look the same. Only 3 d looks like it may be different. Is this a mistake? There are no statistics again to assess what differences are statistically significant, making ecological significance questionable.*

REPLY: We apologize for this error; the right figure is corrected. Statistical analyses are now provided in the manuscript and in the supplementary materials.

[Figure]

Figure 3: Depth profiles of $\delta^{13}C$ for each sampling location (Top-Upper Middle-Middle-Lower Middle-Bottom-Valley) of the four transects including F1 (a), F2 (b, GLA (c) and GLP (d).

**Figure 3.L893**

*L296: Could this be because of deposition from soil that originated upslope via erosion? Could this explain why topsoils don't have more enriched 13C values on the slopes?*

REPLY: We considered all alternatives that might explain the higher value of $\delta^{13}C$ in the valley. We suggested that the $\delta^{13}C$ values of the top position could be due to vegetation changes, which induce higher erosion rates on the upper slope positions. In addition, we observed a higher vegetation density in the valley positions – this has been included in the revised version.

**4.1 Difference in carbon sources between grassland and forest soils. L512-513;L515-516.**

*L355: Rephrase for clarity – something like "Surface erosion is expected to be variable across topographic positions along the hillslope transect, with minimal…."*

REPLY: Thank you for this suggestion; the sentences has been changed as suggested.

**4.1 Difference in carbon sources between grassland and forest soils. L535-L534**

*L357: what does "10Be in-situ topsoil samples" mean? I am more familiar with "in situ 10Be" which means cosmogenic formation of 10Be when surfaces are exposed in rock or sediment. This is an analysis so again this phrasing does not make sense to me. Try "erosion rates from in-situ 10Be analysis of the topsoil samples" perhaps? Also, please clarify what you would expect in terms of variation in the pMC and $^{13}C$ values based on the erosion rates indicated by the $^{10}Be$ analyses.*

REPLY: We apologize for the confusion; the sentences has been modified to clarify. With the data that we have now, we could not really have a specific expectation for pMC and $\delta^{13}C$ values based on the erosion rates indicated by the 10Be analyses. However, the fact that we saw an increase in OC, $\delta^{13}C$ and pMC values at the valley-position under grasslands seems to indicate that at this position soil that has been eroded from upslope position is deposited. This is not observed in the forest transect, which is consistent with low erosion rates, with minimal deposition taking place at the forest valley position.

**4.1 Difference in carbon sources between grassland and forest soils. L528-531**

*L358-9: This is very hard to see in figure 5. It is much easier to see in figure 3 for the $^{13}C$. Please provide a similar figure as Figure 3 to show the 14C value. If it is only useful for this statement, put it in the supplement. I would very much appreciate seeing this figure along with the depth profiles for SOC and 13C as you have shown.*

REPLY: Thank you for this suggestion, we now included a new figure (see below) in the supplementary figures:

[Figure]

Supplementary Figure S6: Depth profile of pMC for  GLP-T, GLP-V, F2-T and F2-V.

**4.1 Difference in carbon sources between grassland and forest soils. L537-538**
**Supplementary Figure S6**

*L360: Again, some statistics would be great and could strengthen your story. Correlation or regression would be very simple but quantify the relationship you see between the isotopes in the grassland transects. On figure 5, drawing a regression line on this plot for the grasslands (and also perhaps for the forests) would also drive home your point about how the grassland values converge with the forest ones at depth and make it easier to identify the depth labels on the different datapoints, which are quite difficult to read. Also, figure 5 would be easier to digest if the grassland points had the same symbols and color, with one open and one closed like the forests. This would make the figure feel less cluttered and make it easier to pick out the labels for the depths and transects. I am unsure why the hillslopes and valleys are marked using different symbols – I do not see a pattern. Is there one? If so this plot should be further improved to make it easier to see. I see the grasslands falling on one regression line and the forests on another.*

REPLY: Thank you for this suggestion. We used different symbols for the valley to show the difference between pMC values found in the valley position of forest and grassland transects. In this figure (Figure 5), we highlighted that OC in the upper 50 cm of the grassland valley profiles is mainly modern (pMC≥100%), which is not the case of the OC in the forest valley position. In addition, we also highlighted that there is a clear trend in the 14C data, whereby the grassland reaches the pMC values of the forest SOC in the subsoil. Therefore, this figure has been improved as suggested, and regression lines have been added:

[Figure]

Figure 1: Comparison of the average value of pMC of bulk SOC plotted against $\delta^{13}$C of SOC of grassland and forest soils. Each point is labelled by the transect position and the top depth interval. Continuous and dotted black lines represent the regression lines of grassland soil samples and forest soil samples, respectively.

**Figure 5. 903**

*L365-378 This section should be significantly shortened to just a few sentences about how your findings are similar to other similar studies. There is no introduction or context about why the stocks or distribution of stocks are important so it is very out of place in a manuscript so focused on vegetation shifts and erosion across hillslopes. What about how similar these soils were prior to when humans may have deforested the current grasslands? What else could explain your results? What about the laterite? What about the lavaka – when did it form and what influence does it have on your findings? What other things may explain your findings other than human deforestation? I very much like the suggestion in this section that there may be long sustained loss of C and I think this is consistent with what long term global evidence for a massive loss of C since the dawn of agriculture has been. But this needs to be better substantiated in the paper through consideration of alternative explanations.*

REPLY: Thank you for this suggestion, as outlined in response to other comments, we have (i) introduced the importance of SOC stocks in the introduction, and (ii) added more context on the laterite horizon depth and lavaka.

**1.Introduction. L96-L106**
**2.1 Study area: Lake Alaotra region. L133-154.**

*Figure 6: Is averaging the grassland profiles like this valid? There is no effect of the lavaka? Are some of these sites influenced by laterite?*

REPLY: We collected all our soil samples on the hillslope and did not find any significant difference between the hillslope with and without lavaka: we have confirmed this statistically in the revised version. The two grassland transects that we analyzed do not show any statistical difference in terms of $\delta^{13}$C and

OC content at the surface, and they show the same trend with depth. We therefore think it is justified to combine the data here.

**2.2 Sampling transect L176-180**
**Supplementary Table 1**

*Figure 7: This is redundant with figure 3, no? chose which one best shows your results (I think figure 3 but it is difficult to tell as it seems to have a mistake). If you like both plot types, move the less impactful one to the supplement.*

REPLY: Figure 3 mainly compares the values in each transect. Figure 7 compares $\delta^{13}C$ of forest and grassland for each position and shows that $\delta^{13}C$ become similar at a lower depth. We agree that the underlying results we present are the same – but feel it is still useful to present them both ways; however we moved Figure 7 to the supplement as suggested.

**Supplementary figure S7**

Response to reviewer #2

*GENERAL COMMENTS*

*I have read with interest this paper, which describes the consequences of vegetation change and erosion processes on SOC dynamic and stocks. It is an interesting research objective, and the purposes of this work would fall within the aims of this journal. In general, I think the paper is interesting and has potential. However the manuscript needs some improvements, outlined in the specific comments, but its main shortcoming is outlined below. The study is based on the comparison of toposequences under forest and grassland and the assumption that the soils under these different vegetations were identical or at least very similar before the vegetation change. However, the paper gives almost no information on these soils, either from a chemical or physical aspect. Some parameters, such as texture, have a strong link with the dynamics and stocks of organic matter. How can we be sure that the very large decreases in C stocks observed under pasture is indeed due to deforestation and the erosion it induces, if we do not know that the soils are really comparable? A presentation of the main characteristics of the soils (if only in the supplementary material) is necessary before we can put forward the hypotheses set out in the discussion. This manuscript, after the necessary improvements and corrections, would be acceptable for publication.*

REPLY: We thank the reviewer for their overall positive evaluation and the detailed suggestions to improve the manuscript. To test our hypothesis whether the differences in SOC and $\delta^{13}$C between grassland and forest profiles are linked to vegetation changes, we agree that additional information on our soil transects would be valuable. Therefore,  additional information on the slope gradient of all transects will be provided in the revised manuscript. We found that slope gradients are similar for all transects, even though the lengths of grassland transects are slightly shorter than the forest ones. We will also include information on soil texture data, which are available and show no significant differences between soils under grasslands and forest. Other specific comments have been addressed point-by-point in our replies below.

*SPECIFIC COMMENTS*

*Abstract*

*Lines 17-18: the time span allowed by the $\delta^{13}$C to study the past dynamic of soil carbon ranges from years to millennia (rather than centuries)*

REPLY: Thank you for this, we  modified "centuries" to millennia as suggested.
**Abstract. L18**

*Line 20: the SOC is low, not extremely low.*

REPLY: Thank you for this, "extremely low"  has been changed to "low".

**Abstract. L20**

*Line 23: "…which show typical profiles under C3 vegetation, with a slight increase with depth."*

REPLY: Thank you for this suggestion, the sentence has been rephrased.

**Abstract. L23**

*Line 30-31: "…suggesting a recent expansion of grass vegetation, and/or that the valleys are depositional areas from organic matter eroded from the hillslopes."*

REPLY: Thank you for this suggestion, the sentence has been changed.

**Abstract. L30-38**

*Lines 31-33: "Our approach, based...determine changing vegetation cover". This is true, but it has already been done in different parts of the world and published in many publications in the last 40 years. As this sentence is written, it sounds like a new approach.*

REPLY: We agree that this approach has been previously applied in different parts of the world, however not yet in Madagascar. We have therefore rephrased this sentence as follows "The method we applied, which is based on the large difference in $\delta^{13}C$ values between the two major photosynthetic pathways (C3 and C4) in (sub)tropical terrestrial environments, provides a relatively straightforward approach to quantitatively determine changing vegetation cover, and we advocate for its broader application across Madagascar to better understand the islands' vegetation history."

**Abstract. L39-42**

*Introduction*

*Lines 87-90: "The stable carbon isotope ratio…show a different degree of isotope fractionation". It is necessary to cite references*

REPLY: The following reference is added: Cerling and Harris (1999).

**Introduction. L119**

*Materials and methods*

*Line 101: the rainfall is not very high; many tropical regions have average annual rainfall between 1500 and 3000 mm or more.*

REPLY: We agree - the term "high" has been removed.

**Materials and Methods . L130**

*Line 103: "the mean annual temperature varies between 18 and 24°C" Really? Not the mean monthly temperature?*

REPLY: We thank the reviewer to notice this, we have corrected this statement accordingly: "The mean annual temperature is 20.6°C, ranging between 11°C in July and 28°C in January (Ferry, 2009)".

**2.1 Study area : Lake Alaotra . L131-L132**

*Line 118-120 AND Figure S3: the length and the gradient of the hillslopes are different under forest and grassland. Could this have an effect on erosion processes?*

REPLY: We agree that the sampled hillslope transects under forest are longer (217 and 184 m) than the sampled grassland profiles (62 and 70 m). However, the slope gradients (derived from the 12 m resolution TanDEM-X DEM) of the four transects are comparable, with maximum slope gradients of 30° and 25° for the forest transects and 29° and 25° under grassland. In the revised manuscript, supplementary Figure S3 has been improved.

[Figure]

S3: Sampling locations along the hillslope transect, plotted together with the elevation profiles (left vertical axis) and slope gradient (right vertical axis). T (Top), UM (Upper middle), M (Middle), LM (Lower Middle), B (Bottom) and V (Valley). Elevation data are extracted from the TanDEM-X DEM.

The two main types of soil erosion on hillslopes are water erosion and diffusive erosion. Water erosion rates typically increase with increasing slope length and gradient (Govers et al., 1994). Diffusive erosion fluxes are approximately proportional to the slope gradient (Heimsath et al., 2005; Pelletier and Rasmussen, 2009; Roering et al., 1999).

Based on the topographical characteristics only (i.e., assuming the same vegetation cover) of our hillslopes, we can thus expect diffusive soil erosion fluxes to be similar for all four transects which would result on lower diffusive erosion rates on the forested slopes as they are longer. Similarly, one might expect higher water erosion rates on the lower half of the forest transects when considering topography only as these are longer than the grassland profiles. However, the effect of slope length on water erosion rates is non-existent under dense, natural vegetation (Cerdan et al., 2010) and it is therefore unlikely that there would be significant differences in erosion rates between the grassland and forest slopes if they would only have a different topography. The differences in erosion rates due to differences in topography are more than likely far less important than those related to differences in vegetation cover. A grass cover that is well below 100% offers far less protection, however: consequently, water erosion rates may be expected to be significantly higher on the grassland slopes in comparison to the forest slopes (Carroll et al., 2000; Silburn et al., 2011).

**Supplementary figure S3**

*Line 123-126: Why is there such a large distance (about 60 km) between the soil profiles under the forest and those under the grassland? Were there no adequate situations for the grassland soils closer to the forest? Important information about the soils is missing, which could be in the supplementary material: are the soils under forest and under grassland really similar, in chemical and physical terms. One of the objectives of the paper is to assess the effect of vegetation change on carbon stocks. Several soil parameters, such as texture, can influence organic matter stocks, so it is important to know whether the soils are similar.*

REPLY: We agree with the reviewer that the distance between the forest and grassland profiles is relatively large. The main rationale behind the site selection was that (i) grasslands on the western side of Lake Alaotra were the main focus, as these represent a large and continuous/homogeneous area with characteristic vegetation cover, for which we hypothesized that vegetation changes (deforestation) may have occurred long enough in the past to result in differences in SOC inventories and characteristics. The nearest zone of pristine forest is located on the eastern side of Lake Alaotra – given the wide alluvial plain that results in a fairly high distance between sites. While grasslands area are also present on the eastern side of the lake, they represent a much more narrow strip of land which may have been deforested relatively recently so that SOC inventories might still reflect the forest cover that was present until recently. However, we paid careful attention to ensure that the topography of the transects was as equivalent as possible.

We agree that the chemical and physical characteristics of our soil should be comparable in order to verify our hypothesis of a shift in vegetation. The soils at both the forested and grassland sampling site are defined as ferralsols (World Reference Base for Soil Resources , 2006). We further verified the assumption of comparable soils by analysing the texture of the soil under forest and grassland. These results are now included, we did not observe significant differences in texture of soils under grassland and forest (p-value =0.663 (sand); p-value=0.723 (silt) and p-value= 0.232 (clay)). In the revised version of the manuscript, we added the soil texture diagram (see below) to the supplementary figures, added a paragraph describing the methods used to derive the soil texture, and report the results of the texture analysis in the text.

[Figure]

Supplementary Figure S5: Texture triangle (clay, silt and sand) of soil under forest and grassland.

**2.2 Sampling transect. L140,**
**2.3.3 Soil texture analyses : L269-L275,**
**3.1 Soil texture for grassland and forest soils. L286-L296,**
**Supplementary figure S5**

*Results*

*Line 207-208: The description of the C profiles is too brief and even wrong! For example, for the F1UM profile the SOC content varies from 60 to 200 cm, between 0.3 and 0.9 %, not 0.1 and 0.2 %.*

REPLY: We apologise for the error. We have verified and corrected these numbers and have further elaborated the description of these results.

**3.2 Organic carbon content and stable carbon isotope of forest soil and vegetation. L298-304**

*Line 210-211: The description of the $\delta^{13}C$ profiles is too brief.*

REPLY: The description of the $\delta^{13}C$ has been extended.

**3.2 Organic carbon content and stable carbon isotope of forest soil and vegetation. L305-L311**

*Line 218-219: It would be better to say that in the first few decimeters, these two profiles have lower SOC values than the other profiles.*

REPLY: Thank you for your suggestion, we have changed this accordingly.

**3.3 Organic carbon content and carbon isotope ratios of grassland soil and vegetation. L316**

*Line 236: The sentence "However, the cumulative…on the GLP hillslope" is unnecessary.*

REPLY: We have removed this.

**3.4 Soil organic carbon stocks for grassland and forest soil profiles. L363**

*Figure 3c: THIS IS NOT THE GOOD ONE!*

REPLY: Thank you for pointing out this error. We corrected the sub-plot, and verified the corresponding text, and checked the full manuscript for correct Figure and Table references.

[Figure]

Figure 2: Depth profiles of δ¹³C for each sampling location (Top-Upper Middle-Middle-Lower Middle-Bottom-Valley) of the four transects including F1 (a), F2 (b, GLA (c) and GLP (d).

**Figure 3.L893**

*Discussion*

*Lines 253-277: All these explanations of the evolution of δ 13C values under C3 forest vegetation are excessively long. Since the end of the 80's, many articles have detailed this. This does not provide decisive information to answer the objectives of the paper.*

REPLY: We had elaborated on this topic to provide the reader with the necessary background information to frame the observed decrease in δ¹³C that we observed under forest, and to be able to properly compare this with the trends we found under grassland that are described from line 278. However, we agree this might be considered too extensive, and reduced the length of this part by removing few sentences in the revised version of the manuscript.

**4.1 Difference in carbon sources between grassland and forest soils. L380-427**

*Lines 293-295: I do not agree, in the topsoil (what depth exactly?), the C3 contribution is much lower than 70%! See the figure 6.*

REPLY: This was indeed not clearly formulated, the profile interval we refer to here is the upper 0-55 cm, and we excluded the values in the valley profile. We clarified this by changing the sentence as follows: " the contribution of C3 plant material to the SOC present in the upper 0-55 cm of these grassland soil profiles is estimated at ca. 70%, with the exception of the valley position."

**4.1 Difference in carbon sources between grassland and forest soils. L444-445**

*Lines 297: for GLP-V the δ13C value increases between the surface and 50 cm.*

REPLY: We have corrected his.

**4.1 Difference in carbon sources between grassland and forest soils. L472**

*Lines 356-358: repetition of the lines 348-350*

REPLY: Thank you for noticing this repetition. Lines 348-350 mainly point out the difference between of erosion which occurred in transects under forest vegetation and grassland vegetation, whereas lines 356-358 refer to differences in erosion between along the transects, i.e. that the erosion rates increase from the top towards the lower slopes. To clarify this, we combined these 2 sections in the new version of the manuscript as follows: " This is confirmed by soil erosion rates derived from in situ $^{10}$Be concentrations of the topsoil samples (5-15 cm) which indicates that both under grassland and forest erosion rates increases from the top towards the valley position, where the erosion rates are consistently higher under grassland when compared to forest."

**4.1 Difference in carbon sources between grassland and forest soils. L528-531**

*Line 380: "…, while the outputs include CO2,…" or "…, while the outputs include CO2 emissions,…"?*

REPLY: Thank you for your clarification. We mean here the $CO_2$ emission, this has been corrected.

**4.2 Response of SOC stocks to vegetation transition. L582**

*Line 397-398: It is not true that all the studies cited found strong differences in SOC stocks between savannah and forest situations. Moreover, the stocks are not calculated and commented on.*

REPLY: We apologize for the confusion due to missing references - we had intended to refer here to Rabetokotany-Rarivoson et al. (2015) and Razafindrakoto et al., (2018) who have investigated the SOC change due to land use change by following the different stages of deforestation that occurred in the humid rainforest of Madagascar. They indeed found that the SOC stocks in the soil under the final stage of deforestation (grasses) are always much lower than the SOC stock under the initial forest. We rephrased this sentences as follow: " Rabetokotany-Rarivoson et al. (2015) and Razafindrakoto et al. (2018) found a strong difference in SOC stocks between the initial forest vegetation and the final stage of deforestation (i.e. a landscape dominated by grasses)".

**4.2 Response of SOC stocks to vegetation transition. L610-611**

*Lines 400-401: That is true, but what does it add to the discussion, at this point. It would be better to delete this sentence.*

REPLY: This has been removed.

**4.2 Response of SOC stocks to vegetation transition. L614**

*Line 411: "The δ13C values of the forest profiles increased with depth, which is expected for soils developed for soils developed under C3 vegetation". It would be better to say that these 13C profiles are typical of soils under C3 vegetation for a very long time.*

REPLY: Thank you for the suggestion. We rephrased this sentence.

**5.Conclusions. L637**

*Lines 417-418: you cannot say that organic carbon input from the new grassland vegetation is not significant: it represents almost a third of the carbon stock!*

REPLY: This description might have been somewhat unfortunate - the fraction of SOC from the grass vegetation indeed represents one third of the total SOC stock. What we aimed to communicate here, is that (i) total OC stocks in the grasslands are substantially lower than in forests, and (ii) that despite the absence of substantial new inputs from C3 vegetation, the bulk of the SOC stocks is still largely dominated (70%) by (old) C3-derived carbon.

To clarify our point, we have rephrased this sentence to make this point more clear and avoid misinterpretations.

**5.Conclusions. L652-654**

*Line 429: "This indicates that the response time to deforestation depends on the rate of depletion of the old C3 pool." What does this sentence mean?*

REPLY: What we referred to here is that the time since deforestation is likely to be reflected in the fraction of the C-OC pool that has been mineralized / lost. We agree that the sentence might be unclear for readers and therefore rephrased this.

**5.Conclusions. L664-665**

*Technical corrections*

*Introduction Line 42: Voarintsoa et al., not Voarintsoa and Cox*

REPLY: Thank you for pointing this out, this has been corrected.

**Introduction. L50**

*Materials and methods Figure 1a: in the caption, it is written "dotted black line", but it is a "dotted white line".*

REPLY: This has been corrected.

**Figure 1. L882**

*Line 121: The supplementary material S3 does not show vegetation*

REPLY: Thank you for pointing out this error; this has been corrected in the revised manuscript.

**2.2 Sampling transect. L172**

*Line 148: in the equation, δ13C, not δ13.*

REPLY: This is now corrected.

**2.3.1 OC content, δ¹³C and ¹⁴C measurements. L212**

*Line 197: for D(i), the unit of measurement is missing.*

REPLY: We added units for D(i) (cm) as well as for the bulk density (g/cm³).

**2.3.2 Dry bulk density and estimation of carbon stock. L262**

*Results*

*Figure 2: in the caption: "middle" not "middles"*

REPLY: We have corrected this.

**Figure 2 caption. L889**

*Line 207: the topsoil samples are 0-5 cm not 0-10 cm*

REPLY: This has been changed.

**3.2 Organic carbon content and stable carbon isotope of forest soil and vegetation. L299**

*Line 209: "…between -25.5 and - 27.1‰ …"*

REPLY: We agree that number format should be one number after the decimal point and it should be -27.1 and -25.5‰ (from low to high values). We changed this in the manuscript and keep our number format consistent.

**3.2 Organic carbon content and stable carbon isotope of forest soil and vegetation. L305**

*Line 225-226: verify the profiles which show gradual decline: GLP-B, GLP-UM, GLA-T (not GLA-M)*

REPLY: It has been verified and changed accordingly in the revised manuscript.

**3.3 Organic carbon content and stable carbon isotope of grassland soil and vegetation. L324-325**

*Line 240: "at different depths" appears two times*

REPLY: This is corrected in the new revised manuscript.

**3.5 ¹⁴C analyses of bulk SOC. L366**

*Line 280: "…values of -20 down…" The symbol ‰ is missing.*

REPLY: The symbol ‰ is now added in the revised manuscript.

**4.1 Difference in carbon sources between grassland and forest soils. L430**

*Line 350: In the references, Brosens et al. is indicated as published in 2022.*

REPLY: The discussed in-situ [10]Be data have not yet been published and are not part of the Brosens et al. (2022) paper. Therefore, we will keep this reference as non-published.

**4.1 Difference in carbon sources between grassland and forest soils. L531**

**References**

Andriamananjara, A., Ranaivoson, N., Razafimbelo, T., Hewson, J., Ramifehiarivo, N., Rasolohery, A., Andrisoa, R. H., Razafindrakoto, M. A., Razafimanantsoa, M. P., Rabetokotany, N. and Razakamanarivo, R. H.: Towards a better understanding of soil organic carbon variation in Madagascar, Eur. J. Soil Sci., 68(6), 930–940, doi:10.1111/ejss.12473, 2017.

Brosens, L., Broothaerts, N., Campforts, B., Jacobs, L., Razanamahandry, V. F., Van Moerbeke, Q., Bouillon, S., Razafimbelo, T., Rafolisy, T. and Govers, G.: Under pressure: Rapid lavaka erosion and floodplain sedimentation in central Madagascar, Sci. Total Environ., 806, 150483, doi:10.1016/j.scitotenv.2021.150483, 2022.

Carroll, C., Merton, L. and Burger, P.: Impact of vegetative cover and slope on runoff, erosion, and water quality for field plots on a range of soil and spoil materials on central Queensland coal mines, Soil Res., 38(2), 313, doi:10.1071/SR99052, 2000.

Cerdan, O., Govers, G., Le Bissonnais, Y., Van Oost, K., Poesen, J., Saby, N., Gobin, A., Vacca, A., Quinton, J., Auerswald, K., Klik, A., Kwaad, F. J. P. M., Raclot, D., Ionita, I., Rejman, J., Rousseva, S., Muxart, T., Roxo, M. J. and Dostal, T.: Rates and spatial variations of soil erosion in Europe: A study based on erosion plot data, Geomorphology, 122(1–2), 167–177, doi:10.1016/j.geomorph.2010.06.011, 2010.

Cerling, T. E. and Harris, J. M.: Carbon isotope fractionation between diet and bioapatite in ungulate mammals and implications for ecological and paleoecological studies, Oecologia, 120(3), 347–363, doi:10.1007/s004420050868, 1999.

Douglass, K., Hixon, S., Wright, H. T., Godfrey, L. R., Crowley, B. E., Manjakahery, B., Rasolondrainy, T., Crossland, Z. and Radimilahy, C.: A critical review of radiocarbon dates clarifies the human settlement of Madagascar, Quat. Sci. Rev., 221, doi:10.1016/j.quascirev.2019.105878, 2019.

Govers, G., Vandaele, K., Desmet, P., Poesen, J. and Bunte, K.: The role of tillage in soil redistribution on hillslopes, Eur. J. Soil Sci., 45(4), 469–478, doi:10.1111/j.1365-2389.1994.tb00532.x, 1994.

Graz, Y., Di-Giovanni, C., Copard, Y., Laggoun-Défarge, F., Boussafir, M., Lallier-Vergès, E., Baillif, P., Perdereau, L. and Simonneau, A.: Quantitative palynofacies analysis as a new tool to study transfers of fossil organic matter in recent terrestrial environments, Int. J. Coal Geol., 84(1), 49–62, doi:10.1016/j.coal.2010.08.006, 2010.

Hackel, J., Vorontsova, M. S., Nanjarisoa, O. P., Hall, R. C., Razanatsoa, J., Malakasi, P. and Besnard, G.: Grass diversification in Madagascar: In situ radiation of two large C 3 shade clades and support for a Miocene to Pliocene origin of C 4 grassy biomes, J. Biogeogr., 45(4), 750–761, doi:10.1111/jbi.13147, 2018.

Heimsath, A. M., Furbish, D. J. and Dietrich, W. E.: The illusion of diffusion: Field evidence for depth-dependent sediment transport, Geology, 33(12), 949, doi:10.1130/G21868.1, 2005.

Mietton, M., Cordier, S., Frechen, M., Dubar, M., Beiner, M. and Andrianaivoarivony, R.: New insights into

the age and formation of the Ankarokaroka lavaka and its associated sandy cover (NW Madagascar, Ankarafantsika natural reserve), Earth Surf. Process. Landforms, n/a-n/a, doi:10.1002/esp.3536, 2014.

Pelletier, J. D. and Rasmussen, C.: Quantifying the climatic and tectonic controls on hillslope steepness and erosion rate, Lithosphere, 1(2), 73–80, doi:10.1130/L3.1, 2009.

Du Puy, D. J. and Moat, J.: A refined classification of the primary vegetation of Madagascar based on the underlying geology: using GIS to map its distribution and to assess its conservation status, Biogéographie de Madagascar, 205–218, 1996.

Rabetokotany-Rarivoson, N., Andriamananjara, A., Razafimbelo, T., Ramifehiarivo, N., Ramboatiana, N., Razafimanantsoa, M., Razafimahatratra, H., Rabeharisoa, L., Bernoux, M., Brossard, M., Albrecht, A., Winowiecki, L., Vagen, T., Grinand, C., Vaudry, R., Rakotoarijaona, J.-R., Rahagalala, P., Rasolohery, A., Parany, L., Bürren, C., Saneho, H. J., Miasa, E. and Razakamanarivo, H.: Changes in soil organic carbon (SOC) stocks after forest conversion in humid ecoregion of Madagascar, XIV WORLD For. Congr. Durban, South Africa, 7-11 Sept. 2015, (September), 8p, 2015.

Razafindrakoto, M., Andriamananjara, A., Razafimbelo, T., Hewson, J., Andrisoa, R. H., Jones, J. P. G., van Meerveld, I., Cameron, A., Ranaivoson, N., Ramifehiarivo, N., Ramboatiana, N., Razafinarivo, R. N. G., Ramananantoandro, T., Rasolohery, A., Razafimanantsoa, M. P., Jourdan, C., Saint-André, L., Rajoelison, G. and Razakamanarivo, H.: Organic Carbon Stocks in all Pools Following Land Cover Change in the Rainforest of Madagascar, Soil Manag. Clim. Chang. Eff. Org. Carbon, Nitrogen Dyn. Greenh. Gas Emiss., (September 2018), 25–37, doi:10.1016/B978-0-12-812128-3.00003-3, 2018.

Riquier, J.: Etude sur les Lavaka, Mem. l'Institut Sci. Madagascar, D(VI), 169–189, 1954.

Roering, J. J., Kirchner, J. W. and Dietrich, W. E.: Evidence for nonlinear, diffusive sediment transport on hillslopes and implications for landscape morphology, Water Resour. Res., 35(3), 853–870, doi:10.1029/1998WR900090, 1999.

Silburn, D. M., Carroll, C., Ciesiolka, C. A. A., DeVoil, R. C. and Burger, P.: Hillslope runoff and erosion on duplex soils in grazing lands in semi-arid central Queensland. I. Influences of cover, slope, and soil, Soil Res., 49(2), 105–117, doi:10.1071/SR09068, 2011.

Voarintsoa, N. R. G., Cox, R., Razanatseheno, M. O. M. and Rakotondrazafy, A. F. M.: Relation between bedrock geology, topography and lavaka distribution in Madagascar, South African J. Geol., 115(2), 225–250, doi:10.2113/gssajg.115.225, 2012.

Wells, N. A. and Andriamihaja, B.: The initiation and growth of gullies in Madagascar: are humans to blame?, Geomorphology, 8(1), 1–46, doi:10.1016/0169-555X(93)90002-J, 1993.

World Reference Base for Soil Resources: A framework for international classification, correlation and communication, World Soil Resources Report 103, FAO, Rome, 2006.

Yoder, A. D. and Nowak, M. D.: Has Vicariance or Dispersal Been the Predominant Biogeographic Force in Madagascar ? Only Time Will Tell, , doi:10.1146/annurev.ecolsys.37.091305.110239, 2006.

---

## Author Response (AR2)

**Responses to reviewers' comments**

We appreciate the positive comments of both reviewers on the revised version of our manuscript. We describe the final changes made below, along with our point-by point response to the reviewers' comments. For clarity, the comments of the reviewers are shown *in italics,* while our response with an indication of the lines and references in the revised MS track-change is presented in normal font.

**Author response Referee #1**

*I highly appreciate the authors' diligence in responding to reviews. This is an interesting data set and study that will be of interest to the scientific community and the revisions have made this a much better paper. I have a few suggestions for minor changes but think this paper is ready for publication after these few changes for grammar and clarity.*

REPLY: First, we would like to express our gratitude to Referee#1 for his/her favorable appraisal and insightful remarks on our revised manuscript.

Referee#1's comments are as follows:

*-Line 42: I think this should be "island's" not "islands'" - as Madagascar is one island.*

Reply : We have corrected this as suggested. (Abstract. L46)

*-L132 should be "...ranging from 11...to 28...." or "ranging between...to..."*

Reply:  We have corrected  this accordingly. (2.1 Study Area. L126)

*-L138" this paragraph is much improved but I wonder if the connection between this historical intensification in grazing and the increase in C4 grasses can be more explicitly stated. If not, perhaps you can present this as a hypothesis that you will support in this study?*

Reply: The relationship between the historical intensification of grazing and the increase of C4 grasses has not been directly explored. There is recent work from our group looking into changes in lavaka erosion rates (Brosens et al. 2022), and work based on pollen and charcoal records in sediment cores of Lake Alaotra that shows a vegetation transition in the catchment (Broothaers et al. in revision). However, neither of these allows us to conclude whether grazing followed an opening in the landscape, or was partially responsible for this shift. As we do not have a clear hypothesis to favor one scenario over another, we prefer to keep this open.

*-L139 "the period considered" seems unfinished (considered for what?) or do you just mean "in this period"?*

Reply: Indeed, we are referring to the period when lavaka erosion rates increased dramatically. Therefore, we modified the text as suggested. (2.1 Study area. L134)

*-L155 "If almost part of…" should be something like "While most of.."I think you can more clearly explain that the C4 grasses are indicative of cattle grazing while C3 grasses and forest vegetation suggest less anthropogenic influence.*

Reply: Thank you for your comments. We have corrected the text as suggested. Indeed, the abundance of C4 grasses could be indicative of the importance of cattle grazing, yet any opening of the landscape would likely result in a dominance of C4 rather than C3 grasses given the temperature & precipitation regime in the region (Zhang et al., 2014). Moreover, in our introduction section, we mentioned that the C4 grasses might be an ancient origin and endemic in Madagascar, thus, in this section (description of the study area) we could not yet assume that the C4 grasses are linked to cattle grazing. (2.1 Study area.L155)

*-L164 "and bamboos" does not flow with the rest of the sentence. Can this be deleted? I am not sure what it means – the forest shade clade C3 grasses are one type and the bamboos are another?*
Reply: We removed "and bamboos" as suggested. (2.1 Study area. L164)

*-L248 it is not entirely clear that the soil samples for 14C analysis were acidified in tin cups because the 14C analysis follows the description of the 13C equations and C3/C4 partitioning. This could be clarified here (e.g. with "on bulk soils loaded into tins and acidified to remove carbonates". Or the description of the analyses for 14C could immediately follow the 13C with the reporting and calculation information provided after.*

Reply : This information was indeed missing in the methods section. The text has been clarified as follows: "Additional $^{14}$C measurements were made at 20–25 cm, 50–55 cm and 100–110 cm depth for GLP-T, GLP-V, F2-T and F2-V (see details on Supplementary Material). These measurements were performed on bulk soil samples which were acidified to remove carbonates". (2.3.1 OC content, $\delta^{13}$C and $^{14}$C measurements. L248-250).

*-L516 change "via the erosion of recent C4 inputs from the upper slopes which are deposited at the valley positions." To "via erosion of recent C4 inputs from the upper slopes and subsequent deposition in the valley". As is there are 2 grammatical issues – "erosion…are" and "which" should be followed by a comma, but I do not think you want to separate your thought this way.*

Reply: Thank you for this correction, the sentence has been modified accordingly. (4.1. Difference in carbon sources between grassland and forest soils. L527-528).

*-L529 is this an increase in erosion rates from the top to the valley or an increase in deposition rates? It seems counter intuitive that erosion rates (soil mass loss) would be higher lower – if this is true can you remind the reader about the shape of your slopes?*

Reply: We do indeed mean erosion rates. While this might seem counter-intuitive, this is related to the convex shape of the slopes, and we have added this aspect to clarify:

"This is confirmed by soil erosion rates derived from in situ 10Be analysis of the topsoil samples (5–15 cm) which indicates that on a convex hillslope, both under grassland and forest, erosion rates increase from the top towards the valley position, where the erosion rates are consistently higher under grassland when compared to forest (L. Brosens et al., unpublished data)." (4.1. Difference in carbon sources between grassland and forest soils. L541).

*-L535 "admixture of young eroded C" – if this is an admixture, what is it mixed with? Do you mean "component" or "addition" rather than mixture?*

Reply: We mean that more significant young eroded C has been added/deposited, hence the sentence has been corrected accordingly. (4.1. Difference in carbon sources between grassland and forest soils. L571)

*-L541 need to specify "grassland soil profiles"*

Reply: Thank you for this remarks, this has been clarified. (4.1. Difference in carbon sources between grassland and forest soils. L577-578).

*-L613 I would use "decades" because often it is expected for this to take multiple decades, not just one and decades is used more commonly than decennia.*

Reply: Thank you for this remark. It has been updated. (4.2. Difference in carbon sources between grassland and forest soils. L635).

*-L620 I like this new paragraph and it flows well from the preceding paragraph. However, it seems out of place. Perhaps it would be better to move the discussion about erosion in the first section (around L528) here? This paragraph is largely about 13C.*

Reply: Thank you for your comment. This paragraph has been moved to the recommended line and section. (4.1. Difference in carbon sources between grassland and forest soils. L544-566)

**Author response Referee #2**

*The authors have made the requested corrections and clarifications. For me the paper is ready for publication after a few technical corrections.*

REPLY: We appreciate and thank referee#2 for his/her favorable evaluation and suggestions.

The comments of Referee#2 are as follows:

*Lines 488-490: "The total SOC stocks in the grasslands are substantially lower than in the forest, and despite the absence of substantial new inputs from C3 vegetation, the bulk of the SOC stocks remains largely dominated (70%) by (old) C3-vegetation". No problem with this sentence, however the first part "The total SOC stocks in the grasslands are substantially lower than in the forest," says the same as lines 485-486. It would be good to rephrase.*

Reply: Thank you for these remarks. Since the sentence "The total SOC stocks in the grasslands are substantially lower than in the forest" has already been stated, we have removed it from the following phrases to avoid this repetition. (Conclusions. L660)

*Line 501: SOC pool, not C-SOC pool*
Reply: Thank you for this remark. It has been rectified. (Conclusions. L672)

**References**
- Brosens, L., Broothaerts, N., Campforts, B., Jacobs, L., Razanamahandry, V. F., Van Moerbeke, Q., Bouillon, S., Razafimbelo, T., Rafolisy, T. and Govers, G.: Under pressure: Rapid lavaka erosion and floodplain sedimentation in central Madagascar, Sci. Total Environ., 806, 150483, doi:10.1016/j.scitotenv.2021.150483, 2022.
- Zhang, Q., Ding, Y., Ma, W., Kang, S., Li, X., Niu, J., Hou, X., Li, X. and Sarula: Grazing primarily drives the relative abundance change of C4 plants in the typical steppe grasslands across households at a regional scale, Rangel. J., 36(6), 565, doi:10.1071/RJ13050, 2014.